Updated or additional text is red. Text that changed position is in green.

# Aerosol composition and the contribution of SOA formation over Mediterranean forests

Evelyn Freney[1], Karine Sellegri[1], Mounir Chrit[2], Kouji Adachi[3], Joel Brito[1], Antoine Waked[1]**,
Agnès Borbon[1], Aurélie Colomb[1], Régis Dupuy[1], Jean-Marc Pichon[1], Laetitia Bouvier[1], Claire Delon[4],
Corine Jambert[4], Pierre Durand[4], Thierry Bourianne[5], Cécile Gaimoz[6], Sylvain Triquet[6], Anaïs Féron[6],
Matthias Beekmann[6], François Dulac[7], and Karine Sartelet[2]

[1]Laboratoire de Météorologie Physique, CNRS-Université Clermont Auvergne, UMR6016, 63117, Clermont Ferrand, France
[2]CEREA, Joint Laboratoire École des Ponts ParisTech – EDF R & D, Université Paris-Est, 77455 Marne la   Vallée, France
[3]Meteorological research institute, Atmospheric Environment and Applied Meteorology Research Department. 1-1 Nagamine, Tsukuba, Ibaraki 305-0052 Japan.
[4]Laboratoire d'Aérologie, CNRS-Université de Toulouse, CNRS, UPS, Toulouse, France
[5]Centre National de Recherches Météorologiques, Météo-France-CNRS, Toulouse, URA1357, France
[6]Laboratoire Interuniversitaire des Systèmes Atmosphériques, LISA/IPSL, UMR CNRS 7583,     Université Paris Est Créteil (UPEC) and Université Paris Diderot (UPD), France
[7]Laboratoire des Sciences du Climat et de l'Environnement, LSCE/IPSL, UMR 8212 CEA-CNRS-UVSQ, Université Paris-Saclay, Gif-sur-Yvette, France
**Now at IMT Lille Douai, Sciences de l'Atmosphère et Génie de l'Environnement (SAGE), F-59508 Douai Cedex, France/Université de Lille, F-59000 Lille, France

*Correspondence to*: *e.freney@opgc.univ-bpclermont.fr*

**Abstract.** As part of the Chemistry-Aerosol Mediterranean Experiment (ChArMEx), a series of aerosol and gas phase measurements were deployed aboard the SAFIRE ATR-42 research aircraft in summer 2014. The present study focuses on the four flights performed in late June early July over two forested regions in the south of France. We combine in situ observations and model simulations to aid in the understanding of secondary organic aerosol (SOA) formation over these forested areas in the Mediterranean and to highlight the role of different gas-phase precursors. The non-refractory particulate species measured by a compact aerosol time of flight mass spectrometer (cToF-AMS) were dominated by organics (60 to 72%) followed by a combined contribution of 25% by ammonia and sulphate aerosols. The contribution from nitrate particles and black carbon (BC) concentrations, were less than 5% of the total $PM_1$ mass concentration. Measurements of non-refractory species from off-line transmission electron microscopy (TEM) showed that particles have different mixing state and that large fractions (35%) of the measured particles were organic aerosol containing C, O, S but without inclusions of crystalline sulphate particles. The organic aerosol measured by the cToF-AMS contained only evidence of oxidised organic aerosol (OOA), without a contribution of fresh primary organic aerosol. Positive matrix factorization (PMF) on the combined organic/inorganic matrices separated the oxidised organic aerosol into a more oxidised organic aerosol (MOOA),

and a less oxidised organic aerosol (LOOA). The MOOA component is associated with inorganics species and had higher contributions of *m/z* 44 than the LOOA factor. The LOOA factor is not associated with inorganic species and correlates well with biogenic volatile organic species measured with a PTR-MS, such as isoprene and its oxidation products (methylvinylketone (MVK), methacroleine (MACR), and isoprene hydroxyhydroperoxides (ISOPOOH)). Despite a

significantly high mixing ratio of isoprene (0.4 to 1.2 ppbV) and its oxidation products (0.2 and 0.8 ppbV), the contribution of specific signatures for isoprene epoxydiols SOA (IEPOX-SOA) within the aerosol organic mass spectrum (*m/z* 53 and *m/z* 82) were very weak, suggesting that the presence of isoprene derived SOA was either too low to be detected by the cToF-AMS, or that SOA was not formed through IEPOX. This was corroborated through simulations performed with the Polyphemus model showing that although 60 to 80% of SOA originated from biogenic precursors, only about 15 to 32% was

related to isoprene (non-IEPOX) SOA, the remainder was 10% sesquiterpenes SOA and 35 to 40% monoterpenes SOA. The model results show that despite the zone of sampling being far from industrial or urban sources, a total contribution of 20 to 34% of the SOA was attributed to purely anthropogenic precursors (aromatics and intermediate/semi volatile compounds).

The measurements obtained during this study allow us to evaluate how biogenic emissions contribute to increasing SOA

concentrations over Mediterranean forested areas. Directly comparing these measurements with the Polyphemus model, provides insight into the SOA formation pathways that are prevailing in these forested areas as well as processes that need to be implemented in future simulations.

## 1 Introduction

The contribution of anthropogenic aerosol particles is thought to be of the order of 10 Tg C yr$^{-1}$, however, that of natural

biogenic aerosols has been estimated to be as much as 90 Tg C yr$^{-1}$, having an important effect on climate in both populated and remote areas of the world (IPCC, 2007, Hallquist et al. 2009). Our knowledge of how primary emissions from anthropogenic and natural sources contribute to the formation of secondary aerosols and their evolution in the atmosphere continues to improve with considerable advances in numerical simulations. However, discrepancies between simulations and measurements still exist and are more apparent over remote and forested environments than over anthropogenic

environments (Ganzeveld et al. 2008; Lelieveld et al. 2008). The most commonly emitted biogenic volatile organic compounds (BVOCs) include isoprene and monoterpenes, with isoprene emissions accounting for approximately 44% (Kesselmeier and Staudt 1999; Arneth et al. 2008). These species can be difficult to characterise because of their high temporal and spatial variability. Studies have shown that the formation yields of SOA from biogenic emissions alone is relatively low compared to those from anthropogenic sources, but when emissions from both biogenic and anthropogenic

sources are combined the resulting yield for SOA formation is much higher than either anthropogenic or biogenic emissions alone  (Day et al. 2009; Bryan et al. 2012; Shilling et al. 2013a; Hu et al. 2015).

The increasing improvement of instrumentation (namely aerosol mass spectrometry) available for the detection of different biogenic species, has led to an increase in the characterisation of biogenic SOA (BSOA), in rural (Schwartz et al. 2010; Slowik et al. 2010; Setyan et al. 2014), Boreal (Kulmala et al., 2000; Allan et al., 2006), Amazonian (Martin et al., 2010), and some other tropical and subtropical forests (Capes et al. 2009; Robinson et al. 2011). Using aerosol mass spectrometry, a

number of studies have identified specific signatures for isoprene derived SOA (Allan et al., 2014; Budisulistiorini et al., 2015). Hu et al. (2015) showed through comparison with model simulations, that the global distribution of a particular SOA formation route from isoprene to epoxydiols is largely focused in the southern hemisphere and over Siberian forests far from anthropogenic emissions. The occurrence of these species in the northern hemisphere has been documented in several studies (Budisulistiorini et al., 2015), but in general the contribution is less than reported in the south hemisphere.

The Mediterranean region is thought to be extremely sensitive to climate change and is influenced by air masses from the Atlantic, continental Europe, and Northern Africa, as well as increasing emissions from biomass burning, intense shipping and from the increasing population density on Mediterranean coastal cities (e.g. Sciare et al. 2003; Barnaba and Gobbi 2004; Lyamani et al. 2006; Alados-Arboledas et al. 2011; Mallet et al. 2013). Several studies have shown that during the summer

months the aerosol radiative effect within the Mediterranean is one of the most significant in the world (Markowitz et al. 2002; Anton et al. 2012; Papadimas et al. 2012). Sartelet et al. (2012) modelled aerosol loading in Europe and North America, and retrieved high concentrations of ozone and SOA over the Mediterranean Sea. It was estimated that biogenic emissions contributed to the formation of up to 72-88% of the SOA over Europe.  In order to better characterise the sources of SOA and its precursors over the Mediterranean, the SAFMED (Secondary Aerosol Formation in the Mediterranean) 

experiment took place in the Mediterranean as part of the Chemistry-Aerosol Mediterranean Experiment (ChArMEx; http://charmex.ipsl.lsce.fr), during Summer 2014. In this work, we present observations from four research flights over the forested Mediterranean region. The objectives of these flights were to combine both aerosol and gas-phase measurements to investigate the origin of SOA over these forested areas.

## 2 Methodology

### 2.1 ATR-42

All airborne measurements were performed aboard the ATR-42 research aircraft, run by SAFIRE (French aircraft service for environmental research; http://www.safire.fr). The ATR (Avion de Transport Régional) is a turbo propeller aircraft of approximately 23 m long and 25 m wide, having a payload of about 4.6 tons (www.atraircraft.com). The aircraft was based in Avignon, France. Aircraft flight plans were decided depending on forecasts from meteorological and air quality models

made available on a dedicated operational web server called the ChArMEx Operation Centre (ChOC; http://choc.sedoo.fr). A series of different standard meteorological parameters were measured aboard the ATR-42 including temperature, pressure, relative humidity, turbulence, wind speed and direction as well as downward and upward radiances.

## 2.2 Online aerosol chemical and physical properties

In order to sample aerosol particle species, a forward facing aerosol inlet was fitted in place of a side window. This inlet is designed with an outer sleeve for channelling air and an inner tube having a large diameter and low curvature to limit particle losses due to deposition. This inlet is both isokinetic and isoaxial and has a 50% sampling efficiency for aerosol

particles with diameters of 4.5 µm (Crumeyrolle et al. 2013). From the aerosol inlet, the sampled aerosols are directed through a manifold to a number of different instruments. Aerosol particle number concentrations were measured using a condensation particle counter (CPC, cut off diameter 5 nm) and scanning mobility particle sizer (SMPS) with 162 size channels for particle diameters ranging from 17 nm up to 400 nm, with a time resolution of 84 seconds. Measurements of aerosol chemical properties were performed using a compact aerosol time of flight mass spectrometer (cToF-AMS)

(Drewnick et al., 2005). This instrument was operating with a time resolution of 40 seconds in order to ensure that the maximum amount of spatial information (aircraft covers approximately 5 km in 40 seconds) could be obtained while maintaining a high enough signal to noise ratio. Prior to being sampled into the cToF-AMS, aerosol particles passed through a pressure controlled inlet (PCI). This inlet maintained a constant pressure of about 400 hPa throughout the duration of the flight and ensured that there were no changes to inlet pressure during in-flight sampling (Bahreini et al., 2008). However, all

reported concentrations are in standard temperature and pressure (used here $22^{o}C$, 950 hPa). In order to provide quantitative information on aerosol mass concentrations, a collection efficiency (CE) must be applied to the aerosol mass concentrations. This is based on the principle that the cToF-AMS aerodynamic inlet is designed to sample dry spherical particles and that particles with non-spherical shapes will not be as efficiently sampled. In addition to this, sampled aerosols particles can sometimes be lost in the instrument as a result of particle bounce on the heating filament. This CE correction is chemical

dependant (Middlebrook et al., 2012), however since the contribution of nitrate and sulphate remained lower than 25% at all times, the CE remained at 50% throughout the sampling period. The total mass measured by the cToF-AMS (added to that from the BC measurements) was compared to the total aerosol concentration measured by the SMPS set up (Fig. S1).  Black carbon concentrations were obtained using a Single particle soot photometer (SP2, Droplet Measurement Technologies (DMT)). Full details of this instrument are available in Baumgardner et al., (2007).

## 2.3 Gas-phase measurements

Gas-phase species were sampled aboard through a rear facing ¼ inch Teflon tube. Ozone and CO were measured using ultra-violet and infra-red analysers (Thermo-environmental instruments) (Nedelec et al. 2003). The NO and $NO_X$ measurements were performed using an ozone chemiluminescence instrument (Environment SA AC42S instrument). The quantification of $NO_2$ is obtained by converting $NO_2$ to NO using a molybdenum converter at 320 °C. As also some $NO_Y$ is converted into

NO in the molybdenum oven, the $NO_2$ and NOx concentration can be overestimated. In this work, these measurements will be referred to as $NO_W$, and represents NO + $NO_2$ + an unquantified NOy. For measurements of volatile organic compounds (VOC), a unit mass resolution proton-transfer-reaction mass spectrometer (PTR-MS) from Ionicon Analytik (Innsbruck,

Austria) was used, with a time resolution of 19s. Full details of the PTR-MS configuration on-board and operating conditions are provided in Borbon et al. (2013). During the so-called biogenic flights, 16 protonated masses were monitored. Compounds of interest are:

- VOCs of biogenic origin (BVOCs) and their first generation oxidation products including $m/z$ 69 (isoprene), $m/z$ 71 (sum of
methylvinylketone (MVK) and methacroleine (MACR) and isoprene hydroxyhydroperoxides (ISOPOOH)), $m/z$ 137 (sum of monoterpenes);

- Anthropogenic volatile organic compounds (AVOCs) including $m/z$ 79 (benzene), $m/z$ 93 (toluene), $m/z$ 107 (C8-aromatics), and $m/z$ 121 (C9-aromatics);

- oxygenated VOCs (OVOCs) including $m/z$ 33 (methanol), $m/z$ 45 (acetaldehyde), and $m/z$ 59 (acetone).

Detection limits, defined as $3\sigma$ of background mixing ratios ranged from 0.05 ppbV to 2.70 ppbV over a 1s dwell time. Instrumental background signal was determined through periodic air sampling (triplicates) of ambient air scrubbed through a custom-built catalyst converter (platinum-coated steel wool) heated to 250ºC. Three complete calibrations over a 0.1-20 ppb range were performed before, during, and after the campaign. The standard gas used was provided by Ionimed (Innsbruck, Austria) and contained several VOCs including isoprene, α-pinene, benzene, toluene and o-xylene at 1 ppmV certified at
±5%. A second ppb-level gaseous standard from NPL (UK) was used to cross-check the quality of the calibration and to perform regular one-point calibration control for isoprene and C6-C9 aromatics (4 ±0.8 ppbV). A relative difference of less than 10% was measured. The calibration factor for all major VOCs (the slope of the mixing ratio with respect to product ion signal normalized to $H_3O^+$) ranged from 2.35 ($m/z$ 137) to 18.9 ($m/z$ 59) counts sec$^{-1}$.

## 2.4 Statistical analysis

Detailed analysis of the organic aerosol mass spectra measured by the cToF-AMS was performed using positive matrix factorization (PMF) (Paatero and Tapper,1994). The PMF2 software package (P.Paatero, University of Helsinki, Finland) was used in conjunction with the PMF evaluation tool (PET) (Version 2.04; Ulbrich et al., 2009). Recommended procedures of down weighting for certain $m/z$ values were followed (Ulbrich et al., 2009) as well as removal of several $m/z$ values due to low ($m/z$ 19 and 20) or high signal ($m/z$ 29). In this particular case, $m/z$ values from inorganic ions ($SO_4$, $NO_3$) were equally
combined with the organic matrices to better separate different factors. Error values for all $m/z$ were calculated in the same way using SQUIRREL software (version 1.53). The number of factors was determined using correlations with external factors (temporal series of VOCs measurements). The reported correlations used later on in the discussion were calculated using simple linear regression.

## 2.5 Electron microscopy analysis

Aerosol particles were collected on transmission electron microscope (TEM) grids using a sampler consisting of two impactor stages. The 50% cut-off of each of these stages was calculated to be 1.6 μm and 0.2 μm respectively, with a flow rate of approximately 1.0 L min$^{-1}$. The samples were collected only when the aircraft was traveling at a constant altitude,

usually lasting between 15 and 20 minutes. The TEM grids on the submicron stage of the impactor were then analysed using a 120 kV TEM (JEM-1400, JEOL) to provide detailed information on individual aerosol compositions and shapes. The advantage of having TEM analysis is the ability to detect both refractory and non-refractory aerosol particles. Composition of each sample collected was analysed using energy-dispersive X-ray spectrometer (EDS) with scanning-TEM mode

(Adachi et al., 2014). There were at least 230 particles analysed from each grid. Particles were classified into five aerosol categories based on their compositions: organic aerosol (C dominant), sulphate (S dominant), sulphate + organic (C and S dominant), sea salt (Na dominant), and other (e.g., mineral dust (Si dominant, but also included traces of Ca and K)). Morphological differences between aerosol particles also allowed us to determine the extent of internal and external mixing. Organic aerosol generally had an amorphous morphology with no evidence of a crystal structure. EDS analysis of the

homogeneously mixed amorphous particles showed that these particles contained C, O, and S without any evidence of crystalline structure. Sulphate particles (likely $(NH_4)_2SO_4$ had a crystalline structure and were sensitive to the electron beam (evaporation). Internal mixtures of organic and sulphate were mostly crystalline $(NH_4)_2SO_4$ surrounded by an amorphous carbon material.

## 2.6 Back trajectory analysis

In order to determine the history of air masses prior to being sampled by the aircraft, air mass trajectories were calculated for a 24-hour period using the Lagrangian model HYSPLIT (http://ready.arl.noaa.gov/HYSPLIT.php). These air mass trajectories were calculated for intervals of five minutes along the flight track and provide information on the dominant air mass sources during the flight. Back trajectories of 24 hours provided enough information to determine whether air masses were slow moving and local or fast moving and arriving over larger distances (Fig. S2). For all flights, the air mass trajectory

path was constant along the flight track at low altitudes, showing that air masses of the same origin were measured during each flight. 72-h back trajectories were also computed (although not shown) in order to determine possible aerosol sources over longer time scales.

## 2.7 Overview of flights

Four research flights (RF) dedicated to biogenic emissions were carried out: the 30[th] of June (RF15), the 3[rd] of July (RF20), the 5[th] of July (RF21), and 7[th] July 2014 (RF23).  Each flight was approximately 3.5 hours in duration, and the aircraft flew over forested areas with elevations varying from 250 to 500 m above ground level (a.g.l) during straight and level runs. The flight plan consisted of the aircraft leaving the city of Avignon (southern France) and travelling east/west for about 50 km before starting a vertical sounding. Vertical soundings were performed from around 100 m up to 3300 m above sea level

(a.s.l). Using the vertical profiles of VOC concentrations and relative humidity, the atmospheric boundary layer height was determined for each flight, and varied from 1300 m a.s.l (RF20) up to 1900 m a.s.l (RF15) (Fig. S3). Two of the flights (RF15, RF21) flew west over the Puéchabon Mediterranean national forest region (North-West of Montpellier, Fig. 1a),

where the principle type of vegetation is evergreen oaks (Quercus ilex) and Alpine pines (halepensis). The evergreen oak is known to emit several different types of monoterpene species but mainly α-pinene (Loreto et al., 1996). The other two flights (RF20, RF23) flew east over the Oak Observatory field site at Observatoire de Haute Provence (O3HP, https://o3hp.obs-hp.fr, Fig. 1b). This area is dominated by downy oak trees (Quercus pubescens) but also contains Montpellier maple (Acer

monspessulanum) and smokey bushes (Cotinus coggygria) in a lower canopy stage. Since Quercus pubescens is the dominant type of vegetation, it makes this region a strong isoprene emitting region and an ideal area to study isoprene chemistry and its relation to aerosol particles (Zannoni et al., 2015).

For the flight RF20, temperatures were stable, varying from 18 to 19$^{o}$C in the boundary layer, and wind speeds were always

less than 5 ±1 m s$^{-1}$, originating from a south-easterly direction. HYSPLIT air mass back trajectories show that for a 24-hour period prior to the measurements, air masses were slow moving and remained within a 200 km radius of the sampling site (Fig. S2b). This, together with the clear skies and relatively high temperatures made ideal conditions to study local biogenic emissions. RF23, had similar temperatures to those recorded on RF20 (17 to 20$^{o}$C), but with some cloud cover. Wind speeds ranged between 2 and 4 m s$^{-1}$, air masses arrived from a southerly direction passing over the coast line prior to being sampled

along the flight track (Fig. S2d). For the two westerly flights, average temperatures were slightly higher 23 ±1$^{o}$C. Wind speeds were low, (3 ±1 ms$^{-1}$). Air masses travelled much greater distances over western (RF15) and northwester (RF21) parts of France prior to being sampled (Fig. S2, a, c).

## 3 Results

### 3.1 Gas-phase properties

The principal VOC species measured by the PTR-MS during all flights were acetone ($m/z$ 59) and methanol ($m/z$ 33), followed by isoprene ($m/z$ 69) and its oxidation products (MVK + MACR + ISOPOOH) ($m/z$ 71) and then VOC species representative of monoterpenes emissions ($m/z$ 137) (Fig.2). Isoprene and its oxidation products showed a high temporal variation during flights suggesting a more local influence of these VOC species. Monoterpene VOCs, having a short atmospheric lifetime were measured in low concentration with little temporal evolution. Anthropogenic VOC species ($m/z$ 93

(toluene), $m/z$ 79 (benzene), and C8- and C9 aromatics) never contributed more than 5% to the total VOC measured (Table 1, Fig. 2). Despite this, we cannot ignore the presence of the anthropogenic VOC species measured during all flights. During westerly flights (RF15 and RF21), air masses arrived from the north (Fig. S2), possibly transporting accumulated anthropogenic emissions from over mainland France. Easterly flights (RF20 and RF23) being principally influenced by local or southerly air masses, are likely impacted by anthropogenic activities over the Marseille and Fos Berre industrial area.

## 3.2 Aerosol chemical properties

In the following section we will report average values for different chemical species measured during low and constant altitude parts of the flights (below the boundary layer height determined from the vertical profiles shown in Fig. S3). For all flights, the aerosol composition measured by the cToF-AMS instrument shows that the organic compounds contributed a

significant fraction to the total aerosol concentration (with average values of 72 (±36) % for RF20 and 71 (±30) % for RF23) (Table 2, Fig. 3). The second most dominant species measured were sulphate and ammonium aerosol particles, with a combined contribution of up to 25 ±10%. The contribution from nitrate species was on average 3 (±1.5) %. Black carbon (BC) measured using the SP2, never exceeded 5% to the total $PM_1$ mass (Fig. 3). O:C values were 1.05 (±0.05) for the total organic aerosol, having high f44 >0.2 and corresponding low f43 <0.6. These mass spectral signatures suggested that the

majority of the organic aerosol was secondary, with little influence from fresh primary organic aerosol.

As described in section 2.5, the chemical composition of aerosol particles collected on TEM grids was determined using EDS. At least 230 particles were analysed during each flight providing information of particle size and composition. The absolute number of particles analyzed using offline electron microscopy is small in comparison to what is measured by

online particle counters, however this technique provides us with a qualitative snap shot into particle mixing state, morphology and composition. Only filters from the submicron stages are discussed here and showed that at least 35 (±5) % of all aerosol particles measured were made up of homogeneously mixed amorphous (no evidence of a crystal structure) particles. EDS analysis showed that these amorphous particles were composed of homogeneously distributed C, O, and S (Fig. 4a i) ii) iii). The molecular structure of these compounds is unknown. Externally mixed crystalline sulphate particles

contributed 15 (±5) % and 10 % were internally mixed amorphous C and crystalline sulphate (likely ammonium sulphate) species (Fig. 4b, Fig S4). The remaining fractions contained signals for sea-salt (Na Cl) and dust (Si, Ca) particles.

## 3.3 Aerosol physical properties

Aerosol number concentrations and size distributions were measured using a CPC and SMPS (respectively) during the four biogenic flights (Fig. 5). During the westerly flight RF15, when air masses were travelling from the north west of France,

particle concentrations were on average 1500 ±300 $cm^{-3}$ and the principle size mode was less than 90 nm at altitudes of around 500 m. At higher altitudes aerosol number concentration decreased to 600 ±200 $cm^{-3}$ with modal diameters of around 30 ±20 nm (Fig. 5a). The measured particle concentrations during the other westerly flight, RF21, were on average 1895 ±1707 $cm^{-3}$. The fraction of fine particles, <40 nm in diameter (F40), measured during these flights was high; explaining the lower aerosol mass measured using the cToF-AMS instrument (Table 2). During the two easterly flights, average aerosol

number concentrations were considerably higher at 3332 ±1920 $cm^{-3}$ than the westerly flights (Fig. 5b and d). The size distribution of the aerosol had a single mode at around 100 nm. However, during periods with increased aerosol concentrations, the size distribution spectra showed an additional mode between 20 and 40 nm (nucleation mode particles).

Calculating the difference in aerosol particle concentrations measured by the CPC (cut off 5 nm), from that measured by the SMPS, we were able to determine the contribution of nucleation mode particles.

The increases in fine mode particles measured at lower altitudes (~500 m. a.g.l) during all flights are likely linked to new particle formation. Observations of new particle formation from biogenic emissions have been reported over Boreal forests (Sihto et al. 2006), European coniferous forests (Held et al. 2004), African savannah forests (Laakso et al. 2013), as well as during laboratory studies (Kiendler-Scharr et al. 2009). Monoterpenes oxidation products were shown to produce new particles by nucleation more efficiently than the isoprene oxidation products (Spracklen et al. 2008; Bonn et al. 2014). Some of these studies have also shown that high concentrations of isoprene relative to monoterpenes can inhibit new particle formation (Kiendler-Scharr et al. 2009; Kanawade et al., 2011), although the underlying processes are not yet clear. Calculating the ratio of isopreneC/monoterpeneC (Carbon associated with Isoprene/Monoterpene) and comparing it to the number concentration of nucleation mode particles (Fig. 6), this relationship between biogenic VOC species and nucleation mode particles was investigated. As a result of the low time resolution of the SMPS, we were limited to a small number of points per flight. Data was combined for all flights, giving average ratios of isopreneC/monoterpeneC varying between 0.05 and 8 (average 3 ±1), with lowest values corresponding to highest fractions of fine particle concentrations. Although the variation among points is high, the general trend of these observations is in agreement with previous field studies over mixed deciduous forests (Kanawade et al., 2011) and with laboratory studies in controlled environments showing that high concentrations of monoterpenes relative to isoprene, can favor new particle formation (Kiendler-Scharr et al., 2009). The average ratios of isopreneC/monoterpeneC measured during these Mediterranean flights were higher than those reported in Finnish forests (ratios of 0.18) and lower than the ones measured in Michigan (ratios of 26.4) and Amazonian forests (ratios of 15.2) (Kanawade et al., 2011). In general, high ratios are associated with a very low or a suppressed number of new particle formation events.

### 3.4 Secondary organic aerosol

From cToF-AMS measurements, average mass concentrations measured during the two easterly flights was approximately 2.0 ±0.5 µg m$^{-3}$, whereas those measured by the westerly flights was considerably less at approximately 1 µg m$^{-3}$, making more detailed analysis of aerosol chemical properties difficult. For this reason the remaining analysis is focused on the two easterly flights.

For both easterly flights, increases in organic aerosol concentrations were observed in the valley area between the two high elevation zones (between 43.6 and 43.8$^{o}$N) during horizontal transects of the flight. For RF20, these increases were accompanied by significant increases in the fine particulate matter between 20 and 40 nm, and also those at 90 nm. In addition, increases in the ratio of isoprene oxidation products to isoprene were observed in the same region, implying a more oxidised air mass. A time series plot of total organic aerosol (OA) with MACR+MVK+ISOPOOH shows a good relationship

(Fig. S5a), and plotting the OA concentration against the ratio of MACR+MVK+ISOPOOH/isoprene provides us with a means to observe the evolution of the organic aerosol with the relative age of the air mass with respect to biogenic emissions (Fig. S5b). The ratios of MACR+MVK+ISOPOOH/isoprene measured during this flight are comparable to those measured over this forested area (0.4 to 0.8) during ground based measurements (Zannoni et al., 2016). We observe a reasonable

correlation (r = 0.46) and positive slope (b = 1.1) with increasing OA as the relative air mass age increases, suggesting that SOA formation is likely to have originated from biogenic precursors. Similar plots were prepared using anthropogenic precursor gases, toluene and benzene (Fig. S6), showing a negative correlation with increasing organic mass concentration of r = 0.35 and a slope of -0.56. However, as the toluene and benzene concentrations are both close to the detection limit, care needs to be taken when interpreting these ratios. Generally, although anthropogenic precursor's species are present, the VOC

concentrations and trends measured suggest that the increases in OA concentrations are primarily related to biogenic emissions.

In order to extract additional information on the OA measured during the flights, we performed PMF analysis. Since the temporal evolution of both the organic and inorganic concentrations was similar, we chose to combine the mass spectral

signatures for $SO_4$ and $NO_3$ into the PMF matrix alongside those of the organic species. Mass spectral signatures of $NH_4$ were not included since higher "noise" was associated with these $m/z$ values. Adding the inorganic signals into the PMF matrix allows us to separate different aerosol sources and not only those related to organic compounds. For both flights a two-factor solution (f-peak 0) was chosen to best describe the sources of the aerosol particles and those 2 factors have maximum correlations with external species (Table 3). Additional details of the PMF analysis are included in Fig S7 to S9 as

well as Table S1. The two resolved factors include (i) a more oxidised organic aerosol (MOOA, contributing 55% to the resolved factors), containing high contributions from $m/z$ 44 and associated with inorganic peaks ($m/z$ 30, 46 ($NO_3$), and 48, 64, 80 ($SO_4$)) and (ii) a less oxidised organic aerosol species (LOOA, contributing 45%) with little contribution from inorganic $m/z$ (Fig. 7a). These two factors MOOA and LOOA are very similar to the OOA-1 and OOA-2 species identified from ground-based measurements during a biogenic event over a forested area in Canada (Slowik et al. 2010). Slowik et al.,

(2010) showed similar trends with the two identified oxidised organic aerosol, where one was associated with inorganic aerosols and the other was not correlated with inorganic aerosols, while the other was well correlated with biogenic VOC species.

During the flights, as the valley area is approached, we observe the sampled air masses become gradually more oxidized

with respect to biogenic emissions, providing us with a well-defined sample area to evaluate the contribution of biogenic SOA on background/regional air masses. In order to isolate the formation of OA resulting from the oxidation of VOC species, the change in the OA concentrations above the background was calculated (ΔOrg). The background values were determined based on measurements during transects of the flight between the valley area and the airport. During this time, aerosol concentrations were low with little temporal variation. Particle size measurements display a single mode at 100 nm

with average particle concentrations of 3000 cm$^{-3}$. Measurements of VOC species during this background period result in average concentrations of Isoprene of 1544 pptV ± 696 pptV, and lower concentrations of longer lived species MACR+MVK+ISOPOOH of 661 pptV ± 239 pptV (Table S2).

For the resolved PMF factors, LOOA and MOOA, background values were determined to be 0.27 and 0.41 μg m$^{-3}$ respectively (Table S2). Organic factors corrected for background concentrations are referred to as Δ-LOOA and Δ-MOOA. Plotting these two factors against the ratio of MACR+MVK+ISOPOOH/isoprene (relative air mass age) (Fig. 7 and S10), we observe a significant increase of the Δ-LOOA species with air mass age until a maximum is reached at ratios of 0.65. Given that MOOA does not change with the relative air mass age in the measured area, and that it is associated with $SO_4$ and $NO_3$

species, it is reasonable to suggest that the MOOA is associated with long-range transported aerosol. A slower increase in concentrations of LOOA at higher ratios suggests that as the relative photochemical age of the air mass increases, LOOA becomes more oxidized and is converted to MOOA, as has recently been illustrated in chamber experiments by Palm et al., (2018). Plotting these two factors as a function of air mass age using anthropogenic VOC species (ratio of toluene/benzene), we observe a relatively flat and decreasing trend (Fig. S10a). These observations would suggest the contribution of toluene

and benzene, although not insignificant play a lesser role in the formation of the SOA measured during these flights.

Given the good correlation between the LOOA and isoprene and its oxidation products, we investigated the possibility to identify isoprene derived SOA. Several recent publications have identified signature peaks in aerosol mass spectrometry for isoprene derived SOA, using the $m/z$ 53 and $m/z$ 82 (ionisation products of IEPOX) (Allan et al., 2014; Budisulistiorini et al,

2015; Zhang et al., 2017). In this study, the contribution of these peaks in both types of spectra was very low (fraction of signal <0.004), although somewhat more pronounced for LOOA. These contributions are similar to background contributions of f82 (fraction of $m/z$ 82 to the total organic signal) observed globally by Hu et al. (2015) and Lee et al., (2016), ranging from 0.0002 and 0.0035, and would lead us to believe that we have no significant contributions of f82 in our aerosol mass spectra. Factors influencing the formation of isoprene SOA include aerosol acidity and the presence of $NO_X$

and sulphate (Nguyen et al., 2014), with highest yields of isoprene SOA being measured under low-$NO_X$ conditions (<30 pptV) and in the presence of acidic aerosols (Gaston et al., 2014). Aerosol concentrations measured by the cToF-AMS appear to be fully neutralised with little evidence of acidity (Fig. S11), and the $NO_W$ concentrations measured during these flights varied from 6 up to 10 ppbV, however the average concentrations of NO were 0.30 ±0.2 ppbV, suggesting that the real contribution of $NO_X$ (See section 2.3) is also likely to be low, but still higher than pptV level  concentrations measured

in truly remote forested areas. There have been some reports of isoprene-derived SOA formation (hereafter isoprene SOA) in high-NO regions but the contribution of this pathway is considered to be much smaller (Jacobs et al., 2014).

Other sources of biogenic SOA can originate from the oxidation of monoterpene and sequesterpene VOC species, or additionally from isoprene SOA, that do not follow the IEPOX route. In both cases, the contribution of m/z 91 in the cToF-

AMS mass spectra, often identified as being the $C_7H_7^+$ fragment (Lee et al. 2016; Riva et al., 2016) would be enhanced. This *m/z* 91 was present in all OA mass spectra and was higher for the LOOA (f91 = 0.007). However, in previous studies these f91 values are considered as background (Hu et al., 2015 Lee et al., 2016), hence making it difficult to associate the measured SOA with these formation routes. It should be noted that *m/z* 91 can also be associated with fragments of primary anthropogenic OA, and the contribution of anthropogenic aerosols from the industrial zone (Fos sur Mer) south of the flight area cannot be ruled out.

In general, the yield of formation of SOA from isoprene VOC precursor is relatively low compared to other biogenic species such as monoterpenes, and also compared with aromatics precursors (Ait-Helal et al., 2014). Since the measured aerosol particles are neutralized (Fig. S11) and the measured NO concentrations are still reasonable high (0.30 ppbV), we assume that isoprene derived SOA, following the IEPOX formation route, do not contribute significant amounts to the OA measured during these flights. Given the increase in OA with the relative "biogenic" air mass age, we could suspect that additional sources of SOA could originate from other isoprene SOA formation routes and/or terpene precursors. This is also coherent with the increase in the number concentrations of fine particles at lower IsopreneC/monoterpeneC ratios discussed in section 3.3.

### 3.5 Model evaluation of secondary organic aerosol formation

In order to evaluate the relative contribution of the different gaseous precursors to SOA formation over these forested regions, two simulations were performed using the Polyphemus model. Full details of the model set up are available in Chrit et al., (2017). The domain of the air-quality simulation has an horizontal resolution of 0.125° x 0.125°, while the vertical is modelled with 14 layers having interface heights at 0, 30, 60, 100, 150, 200, 300, 500, 750, 1000, 1500, 2400, 3500, 6000, 12000 m. above sea level (a.s.l) (Fig. S12). Biogenic emissions are computed using MEGAN (Guenther et al. 2006), and anthropogenic emissions using HTAP-v2 (http://edgar.jrc.ec.europa.eu/htap_v2/). Initial and boundary conditions are obtained from a larger-scale simulation (over Europe), as detailed in Chrit et al. (this special issue). For gaseous chemistry, a carbon bound approach model is used (CB05; Yarwood et al., 2005). Aerosol dynamics is modelled with a sectional approach (SIREAM; Debry et al., 2007), and for SOA modelling, a surrogate approach is used (Couvidat et al. 2012). The modelling of SOA formation is based on smog chamber experiments, which provide information on SOA yield as a function of organic mass concentration for each precursor using an Odum approach (Odum et al., 1996). Stoichiometric coefficients of SOA surrogates and their saturation vapor pressures are selected to fit data from smog chambers. Candidates for SOA surrogates are estimated from the literature (Couvidat et al. 2012). Biogenic precursors are isoprene, monoterpenes (with α-pinene and limonene as surrogates) and sesquiterpenes, with low-NOx and high NOx oxidation regimes. Isoprene may form two surrogates, amongst which methyl methyl dihydroxy dihydroperoxide under low NOx, and methyl glyceric acids under high NOx. Monoterpenes may form pinonaldehyde, norpinic acid, pinic acid, 3-methyl-1, 2, 3-butanetricarboxylic acid (MBTCA) under low-NOx, and organic nitrate, as well as extremely low volatile organic carbons (ELVOCs) or highly

oxidized multifunctional organic compounds (HOM) by ozonolysis. Anthropogenic precursors are toluene, xylene and intermediate/semi volatile organic compounds (I/SVOC). Gas-phase emissions of I/SVOC are estimated by multiplying emissions of primary organic aerosols by a factor 1.5 (Zhu et al. 2016). Partitioning between the gas and aerosol phases is done with secondary organic aerosol processor model (SOAP) (Couvidat and Sartelet, 2015) for organics and inorganic

aerosol model ISORROPIA for inorganics (Nenes et al. 1998). Maps of the simulated submicron organic matter ($OA_1$) are shown in Fig. S12a and S12b for the two easterly flights RF20 and RF23 flights, respectively.

Figure 8 compares the vertical profiles of measured and modelled $OA_1$ during the RF20 and the RF23 flights. The concentrations averaged over the vertical layers of the model and the standard deviations around the mean concentrations are

shown. The measured concentrations have higher standard deviations than the modelled concentrations, due to the coarse horizontal model resolution (0.125° x 0.125°). For both flights there are some differences between the model and the measurements.  However, this discrepancy may be due to difficulties in representing the vertical distribution of pollutants above the canopy. Although the mean vertical concentrations of $OA_1$ tends to be under-estimated over 1000 m, they are on average well modelled under 1000 m within the boundary layer for both flights.

Although isoprene emissions are 2.5 times higher than those of monoterpenes and 11.6 times higher than those of sesquiterpenes over that region during the period of simulation, isoprene-derived SOA represents about 15 to 35% of the simulated OA, which is lower than the monoterpene-derived SOA that represents 35% to 40%. Sesquiterpenes-derived SOA represents about 10%. Amongst those monoterpenes-derived SOA, 4% to 7% are monoterpene products (first generation

semi-volatile organic compounds: pinic acid, norpinic acid and pinonaldehyde), 9% to 14% are ELVOCs/HOMS, and 17% to 23% are organic nitrate. In total, biogenic-derived OA represents about 66% to 80% of OA. The rest is made up by aromatics derived OA (2% to 3%) and anthropogenic intermediate and semi-volatile organic compounds (17% to 31%). The contribution of organic nitrate modelled is not reflected in the measurements where less than 5% of the total measured mass was nitrate aerosol. This difference may be due to hydrolysis not being accounted for in the model. Under ambient

conditions hydrolysis could eliminate the organic nitrate functionality, allowing nitric acid to evaporate from the particles (Rindelaub et al., 2016).

The measured organic matter is highly oxidized during both flights with an average measured ratio O:C below 1000 m of 1.05 during the RF20 flight and 1.1 during the RF23 flight. This ratio is very well represented by the model with an average

values of 1.07 (RF20) and 1.17 (RF23). In the model, these high O:C ratios arise because of organic compounds from isoprene oxidation, which all have O:C ratio greater than 0.8, as well as some ELVOCs compounds (monomers) from monoterpenes oxidation. We can conclude from these observations that the low volatility products (ELVOCs) from monoterpenes oxidation as well as isoprene oxidation products may therefore correspond to the measured LOOA concentrations. Although the SOA contribution of anthropogenic VOC precursors is low (Couvidat et al, 2013; Sartelet et al.

2018), the results of the model show a high contribution of anthropogenic compounds (up to 30%). These anthropogenic compounds could correspond to the regionally transported SOA, potentially identified as MOOA.

## 4 Conclusion

This paper characterises aerosol and gas-phase physical and chemical properties over two different forested areas in southern France. During four dedicated flights, aerosol particle and gas-phase composition were measured using a cToF-AMS and a PTR-MS, respectively with the principle objective to characterise biogenic emissions. Aerosol particle physical properties were measured using a number of different techniques characterising particle size and number concentrations. Using a combination of aerosol size distributions coupled with VOC concentrations we observe that although new particle formation seem to occur over all types of vegetation (mainly isoprene-emitting species or mainly monoterpene-emitting species), that low isopreneC/monoterpeneC ratio can favour the formation of fine aerosol particles. These VOC species are likely condensing on pre-existing particles that can then be chemically analysed.

During eastern flights, in valley areas, high concentrations of organic aerosol and biogenic VOC species were measured (isoprene and its oxidation products MACR+MVK+ISOPOOH). PMF analysis of the organic mass spectra separated two organic factors namely a more oxidised organic aerosol (MOOA) and a less oxidised organic aerosol (LOOA). The MOOA species were strongly associated with $SO_4$ species whereas the LOOA species were not related to inorganic species but correlated with temporal evolutions of biogenic oxidation products (MACR+MVK+ISOPOOH). Correlation with other precursor biogenic or aromatic VOC species was very weak.

A lack of direct evidence of IEPOX SOA ($m/z$ 82 $C_5H_6O^+$) in the cToF-AMS measurements leads us to conclude that the formation of SOA, following an IEPOX formation route from isoprene precursor species was not dominant during this measurement period. The Polyphemus model determines a contribution of isoprene SOA, formed through alternative pathways, of the order of 15 to 35%. Therefore, although not possible to accurately identify the formation pathway of the measured SOA, we can, based on its correlation with the oxidation products of isoprene, propose that it is at least partly associated with biogenic isoprene VOC species. The model also illustrates that although the emission of monoterpene and sesquiterpene species is low compared to that of isoprene, the yield of SOA formation from these precursor species is important. This is in agreement with recent observations by Zhang et al., (2018), who showed that SOA is principally formed from monoterpene emissions in southern USA

The model results estimates an overall contribution of 66% biogenic species and approximately 30% anthropogenic influence to the formation of SOA. The model can successfully replicate the measured OA during the flights, as well as the OA oxidation properties. However, the detail molecular information obtained in the model (isoprene SOA, monoterpene

SOA, anthropogenic, organic nitrate) was not easily comparable to measurements. The model resolved organic nitrate contributions up to 17 to 23 %. This high contribution of organic nitrate is not reflected in the cToF-AMS measurements where nitrate contributed to less than 5% to the measured PM1 mass. This difference is possibly due to nitrate hydrolysis that is not considered in the mode.

This study takes advantage of measurements sampling regional air masses that were gradually enriched with biogenic compounds, allowing us to evaluate the contribution of biogenic SOA in ambient environments.  These measurements are compared directly with model simulations, highlighting that there are several atmospheric processes that cannot be neglected by atmospheric models (e.g contribution of ELVOC), as well as emphasizing important processes that need to be

implemented into future model simulations (e.g hydrolysis of organic nitrates).

## Acknowledgements

This study received financial support from MISTRALS by ADEME, CEA, INSU, and project Meteo-France. This research was also funded by the SAFMED (Secondary aerosol formation in the Mediterranean) ANR, (grant number: ANR-12-BS06-

0013-2502). The authors would like to extend a special thanks to the pilots and flight crew from SAFIRE for all their enthusiasm and support during the measurement campaign aboard the ATR-42 aircraft. In addition, the authors are very grateful to Eric Hamonou for his logistical help in organizing the campaign, and to Laurence Fleury, Hélène Ferré and the OMP/SEDOO team who provided excellent support to aircraft operations through the ChOC web interface setup and management.

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

**Table 1. Mean concentrations of the different gas phase species measured during low and constant altitude of each flight. The error represents ±1σ on all the measurements.**

| Flight | dd/mm (2014) | Isoprene (pptV) | MVK+ **MACR+** ISOPOOH( pptV) | Monoterpe nes (pptV) | Toluene (pptV) | Benzene (pptV) | C8+C9 aromatics (pptV) | O₃ (ppbV) | CO (ppbV) | NO_W (ppbV) | NO (ppbV) |
|---|---|---|---|---|---|---|---|---|---|---|---|
| **RF15** | 30/06 | 583 ±290 | 214 ±91 | 117 ±82 | 146 ±59 | 93 ±61 | 200 ±85 | 40 ±8.8 | 118 ±27 | 4.2 ±0.8 | 0.17 ±0.3 |
| **RF 20** | 03/07 | 1240 ±527 | 756 ±287 | 205 ±107 | 149 ±82 | 102 ±42 | 196 ±79 | 53 ±4.0 | 136 ±46 | 7.9 ±2.3 | 0.31 ±0.2 |
| **RF 21** | 05/07 | 600 ±262 | 365 ±182 | 179 ±128 | 147 ±176 | 108 ±53 | 135 ±39 | 31 ±8.0 | 79 ±11 | 5.86 ±0.7 | 0.29 ±0.3 |
| **RF 23** | 07/07 | 392 ±197 | 230 ±159 | 119 ±87 | 74 ±34 | 88 ±18 | 125 ±36 | 52 ±3.0 | 88 ±9 | 5.6 ±2.1 | 0.29 ±0.9 |

5    **Table 2. Concentrations (µg m⁻³) of the different chemical species measured aboard each flight during level low altitude legs., error values are standard deviations calculated on the mean values of the measurements.**

| Flight | dd/mm (2014) | NR-PM1 µg m⁻³ | Org µg m⁻³ | SO4 µg m⁻³ | NO3 µg m⁻³ | BC µg m⁻³ |
|---|---|---|---|---|---|---|
| **RF15** | 30/06 | 0.70 ±0.08 | 0.48 ±0.23 | 0.08 ±0.07 | 0.013 ±0.01 | 0.03 ±0.02 |
| **RF 20** | 03/07 | 2.70 ±1.10 | 1.79 ±0.70 | 0.48 ±0.2 | 0.10 ±0.06 | 0.11 ±0.03 |
| **RF 21** | 05/07 | 0.99 ±0.50 | 0.72 ±0.40 | 0.20 ±0.09 | 0.03 ±0.02 | 0.11 ±0.04 |
| **RF 23** | 07/07 | 1.64 ±0.70 | 0.96 ±0.60 | 0.30 ±0.18 | 0.06 ±0.07 | 0.08 ±0.06 |

**Table 3: Pearsons r (Pr) correlations for different time series during RF 20  and RF 23.**

|  | MOOA | | LOOA | |
| --- | --- | --- | --- | --- |
|  | RF 20 | RF 23 | RF 20 | RF 23 |
|  | Pr ( n =91) | Pr (n= 144) | Pr (n =91) | Pr (n =169) |
| PTR-MS *m/z* 69 (Isoprene) | 0.11 | 0.41 | 0.51 | 0.67 |
| PTR-MS *m/z* 71 (MVK+MACR+ISOPOOH) | 0 | 0.28 | 0.64 | 0.71 |
| PTR-MS  *m/z* 137 | 0.59 | 0.58 | 0.29 | 0.49 |
| PTR-MS *m/z* 93 | 0.15 | 0.25 | 0.27 | 0.38 |
| PTR-MS *m/z* 79 | 0.04 | 0.21 | 0 | 0.15 |
| BC | 0.52 | 0.48 | 0.61 | 0.48 |
| CO | 0.37 | 0.48 | 0.34 | 0.60 |
| NO$_W$ | 0.56 | 0.44 | 0.51 | 0.60 |

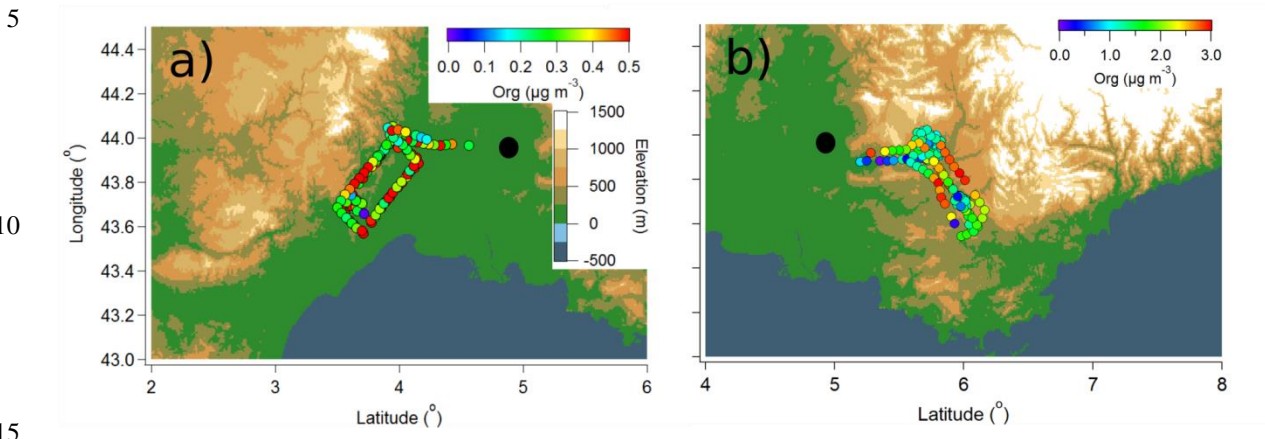

**Figure 1. Typical flight track traveling a) west (RF 15 and RF 21) and b) east (RF 20 and RF 23) of Avignon (black circle). Points of the flight track are colored by organic aerosol concentrations.**

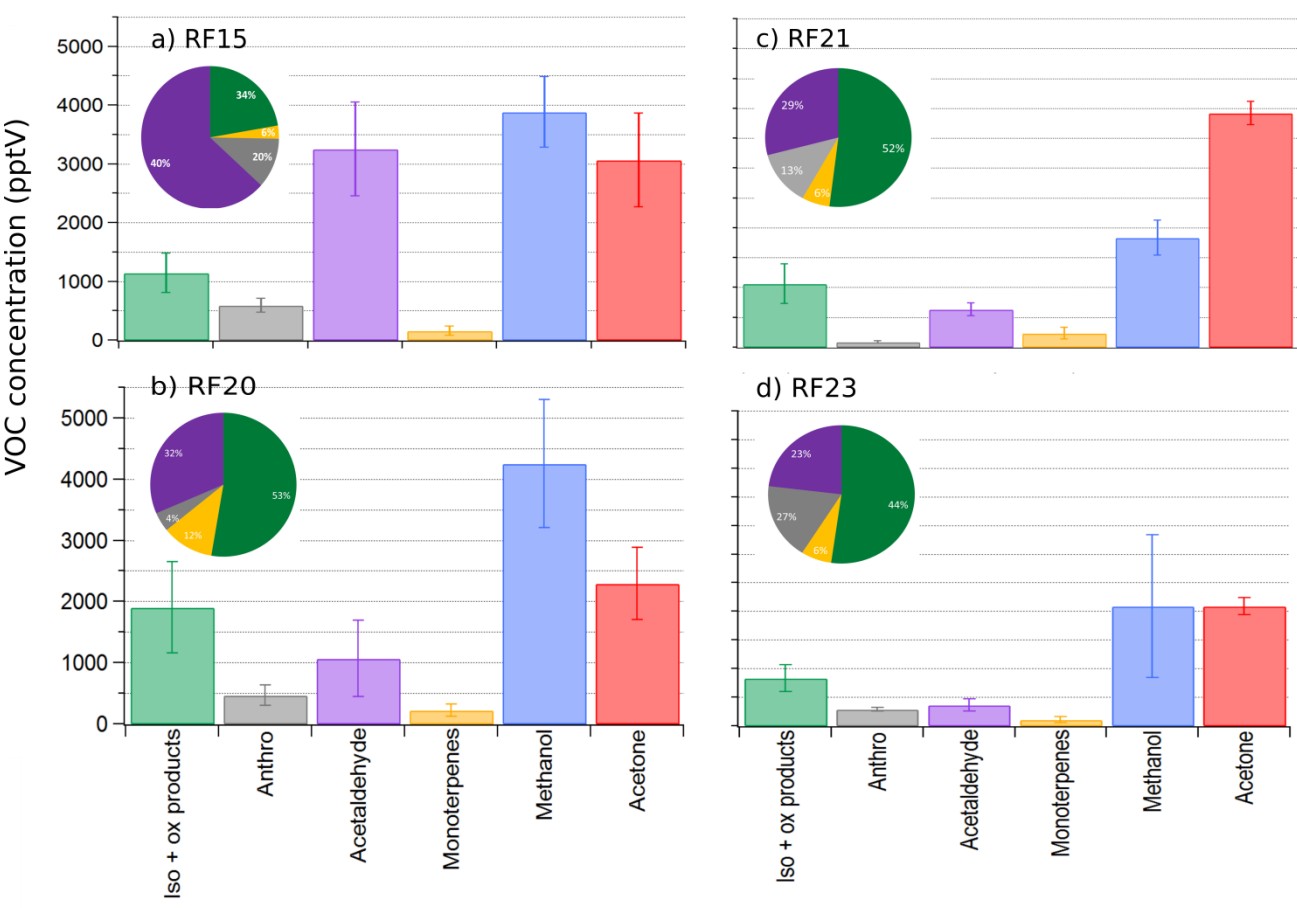

**Figure 2. Contribution of the different measured gas phase species aboard the ATR-42: a) RF15, b) RF20, c) RF21, d) RF23. The pie charts illustrated in each figure represent the contribution of all VOC species *except those of methanol and acetone*.**

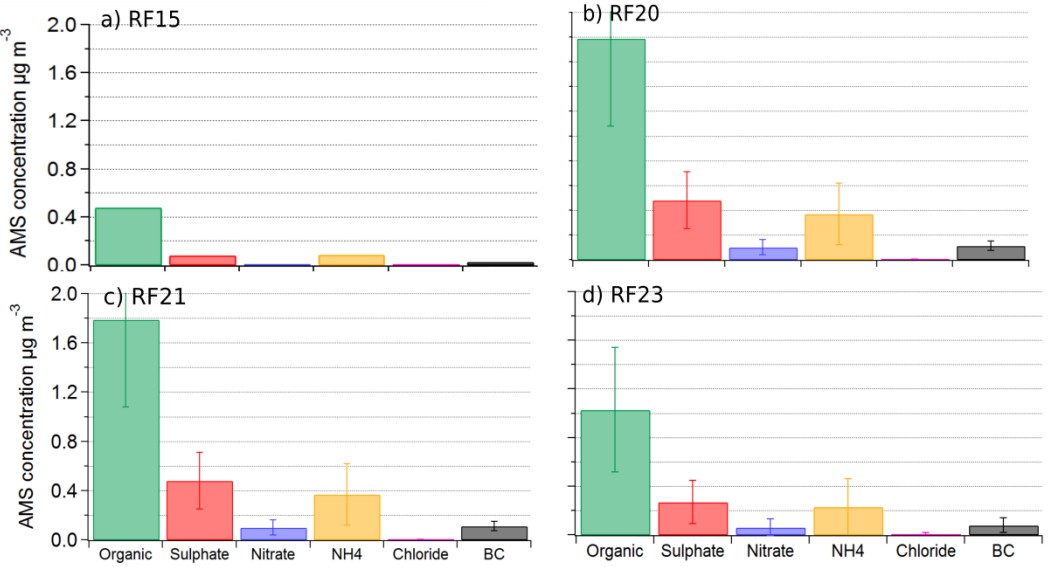

**Figure 3. Contribution of the non-refractory aerosol chemical species aboard the ATR-42: a) RF15 3006 b) RF20 0307, c) RF21 0507, d) RF23 0707.**

a) Homogeneously mixed amorphous organic aerosol particle

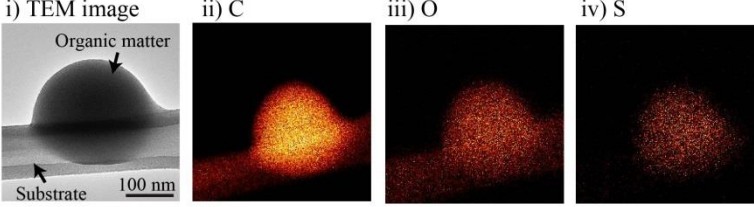

b) Internally mixed organic aerosol particle with sulfate

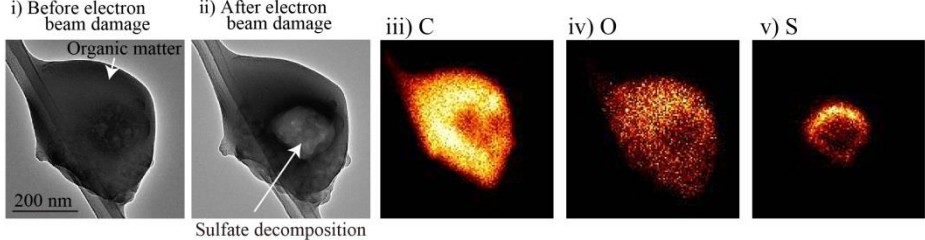

5   Figure 4. a) i) Example of an amorphous particle deposited on a carbon substrate. EDS mapping analysis showing signals for ii) C Carbon, iii) O Oxygen and iv) S Sulfur.; b) Internally mixed amorphous particles with signals, i) before and ii) after electron beam damage. EDS analysis showing signals for iii) C carbon, iv) O Oxygen, and v) S Sulfur.

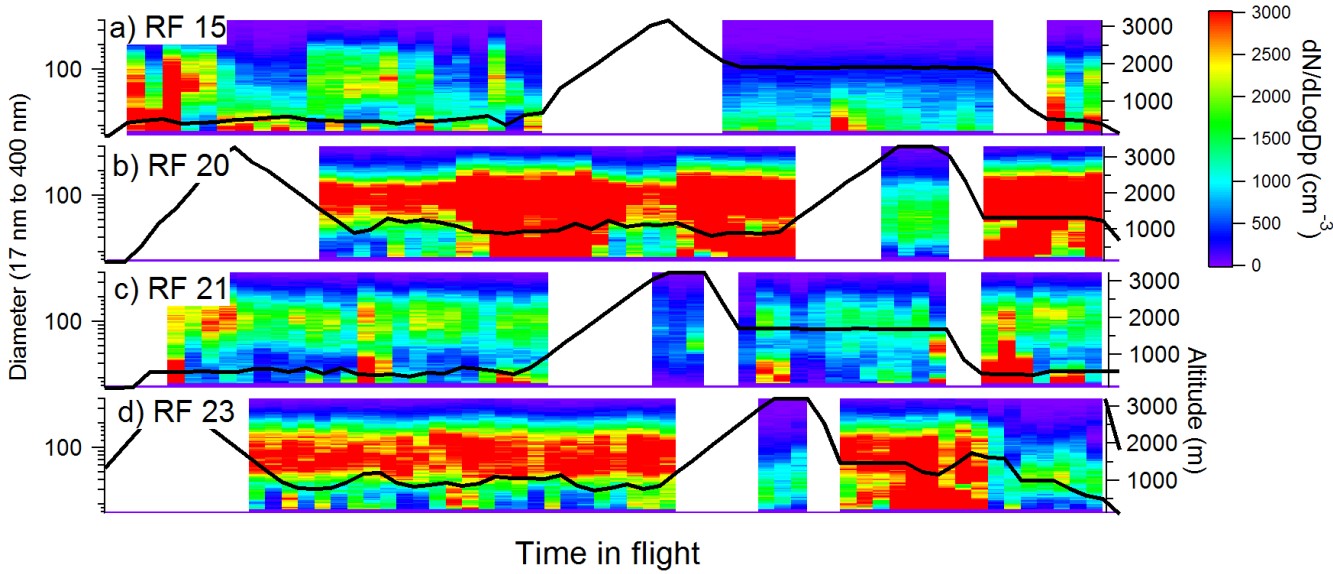

**Figure 5. Aerosol size distribution measured by the SMPS for a) RF15 b) RF20, c) RF21, d) RF23 from 17 nm up to 400 nm. The color scale indicates aerosol concentration dN/dlogDp. Altitude is illustrated as the black line and is represented on the right hand axis.**

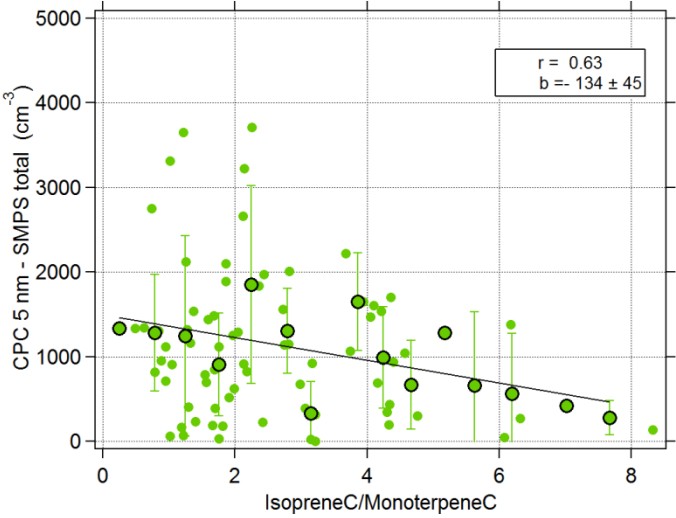

**Figure 6. Ratios of IsopreneC/MonoterpeneC plotted as a function of the nucleation mode particles (difference between the CPC (cut off 5 nm) and the SMPS (cut off 17 nm)). Values for the four biogenic flights are included, as well as average values calculated over a number of IsopreneC/MonoterpeneC ratios (size bins of 0.5); Error bars represent ±1 σ of the average CPC5nm – SMPS values. . The black line represents the linear correlation fit.**

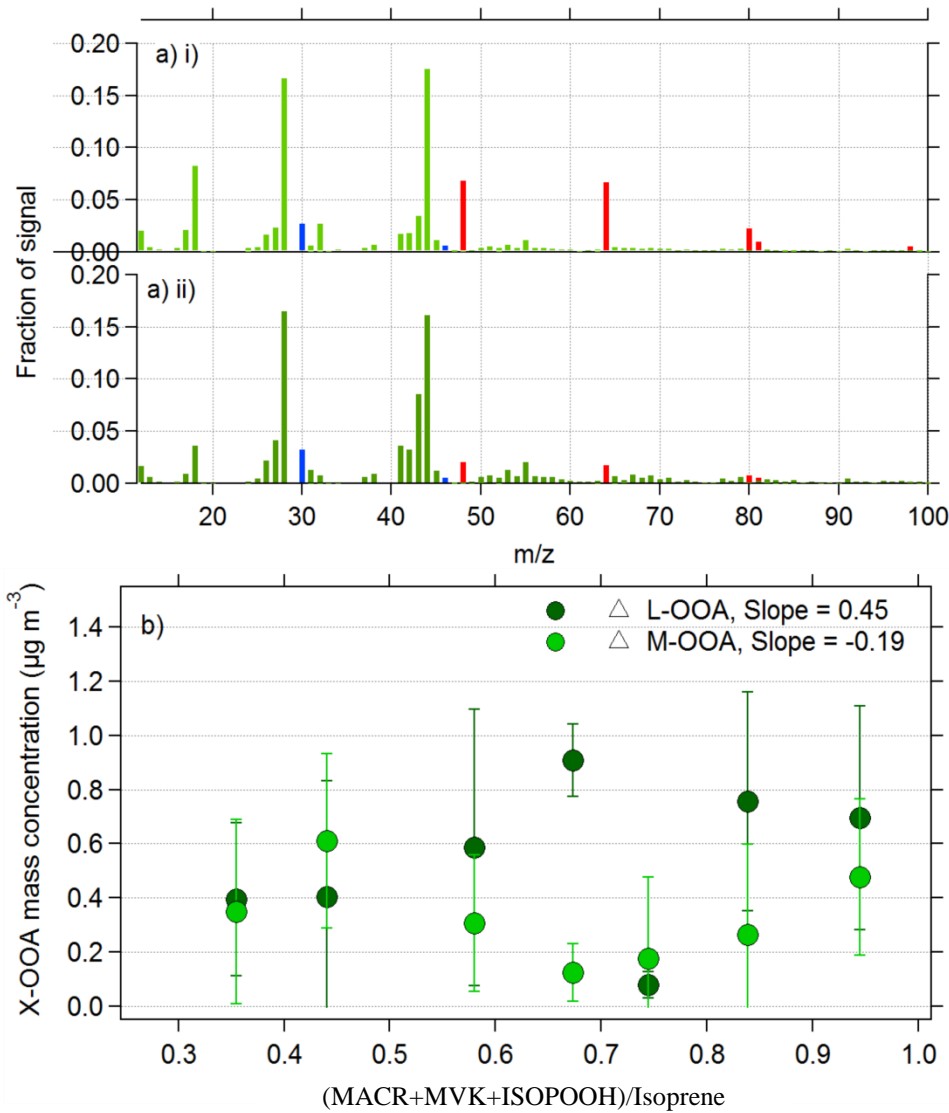

**Figure 7. a) A two factor solution determined from PMF analysis of the biogenic research flights. A) i) The more oxidized organic aerosol (MOOA) associated with inorganic peaks for sulphate (red) and nitrate (blue), a) ii) the less oxidized organic aerosol (LOOA) with a lower contribution of inorganic peaks. b) Variations of these two species with aging air mass (using MACR+MVK+ISOPOOH as a proxy for photochemical age of air mass). The delta values ($\Delta$) are calculated based on background concentrations measured outside of the study region.**

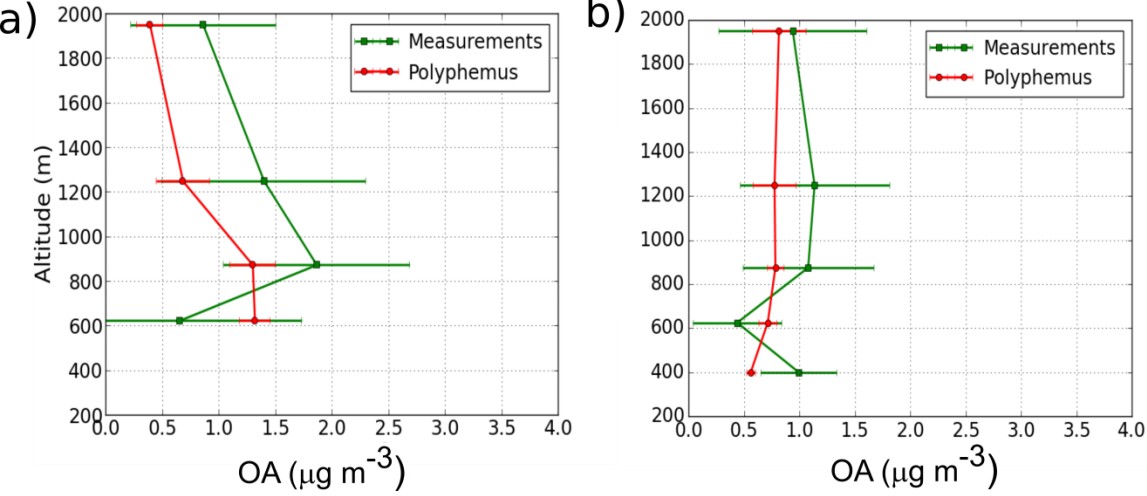

**Figure 8. Measured (green) and modelled (red) organic concentration during the a) RF20 and the b) RF23 flights. The concentrations are averaged on the vertical layers of the model and variations around the average are indicated by the horizontal error bars.**

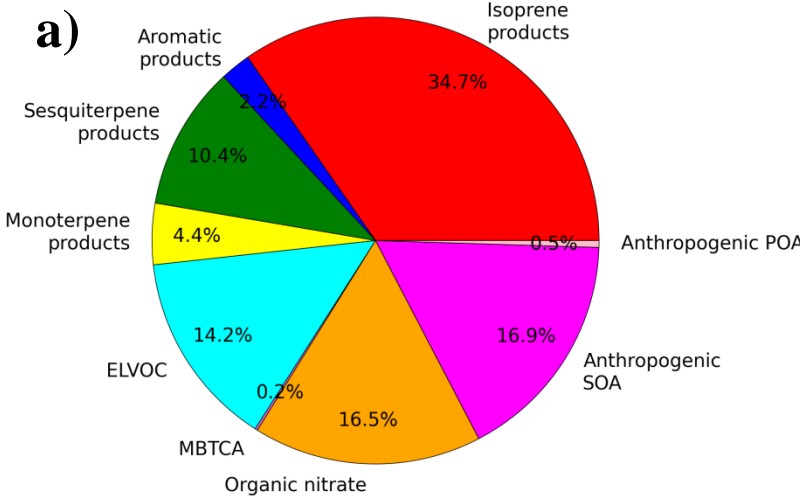

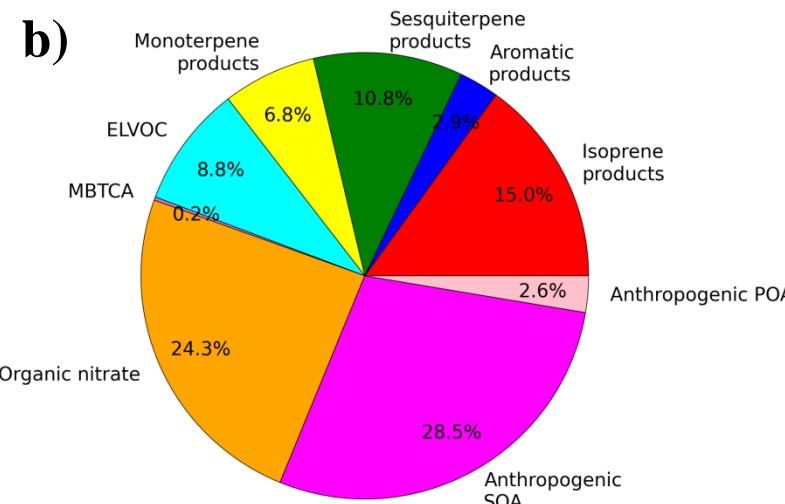

**Figure 9: Modelled averaged composition of OA$_1$ along the flight path during the a) RF20 and the b) RF23 flights. This averaged**
5    **composition is obtained by averaging concentrations along the flight path at altitudes below 1000m.**