# Peer review of "Aerosol composition and the contribution of SOA formation over Mediterranean forests"

_Atmospheric Chemistry and Physics, 2017_

## Referee Comment (RC1) · Anonymous Referee #3 · 14 Sep 2017

Authors present the results obtained during an airborne measurement campaigns over 2 Mediterranean forested areas (South of France). More precisely, they report results combining c-ToF-AMS, HS-PTR-MS, SP2, SMPS/OPC, Aerolaser (for formaldehyde measurements) as well as offline TEM analysis, from 4 flights (2 for each forest). Simulations performed with the Polyphemus model and comparisons with the measurements are also presented. While the paper is well written with a clear structuration, significant work is still needed to make it suitable for publication in ACP. Important details are missing and some major issues can also be found here and there throughout the manuscript (despite 21 co-authors).

Main comments:

1- There are major inconsistencies between VOCs concentrations reported in the text

(p2 line 6, for instance), in figure 1 and in table 1. Not only a question of units. Must be checked very carefully and corrected.

2- Regarding PMF analysis, much more details are needed (at least in the SI)

3- It's not clear if the average values reported in the text and tables refers to the whole flight or only to the low altitude legs. This should be clarified and homogenized throughout the text. Considering the vertical profiles presented in figure S4, whole flight averages make no real sense. These vertical profiles and their implications are not discussed in the text except for boundary layer heights (p6, line 20).

4- Either the definition of externally mixed aerosol has recently changed either I don't understand the difference between fig 3a and b. This leads to important confusions in the discussion and conclusions.

Other comments

P1 line 30-31. Direct comparisons between AMS and TEM analysis are not relevant. AMS refers to a mass concentration without distinction between internally and externally mixed aerosol particles and TEM analysis refers, at best, to a number concentration. Authors state that at least 50 particles per gird have been analysis. What is the representativeness of the analyzed particles with respect to the total particles population? (SMPS/OPC measurement can help to assess this representativeness)

P 1 line 32. "Externally mixed organic aerosols, were equally identified with S signals, which may suggest the presence of organo-sulphates" I really don't understand this sentence. (cf main comment 4)

P2 line 6 : "high mixing ratio of isoprene (2-4 ppbV) and oxidation products (0.6 and 1.2 ppbV)" These concentration ranges cannot be found in table 1 and figure 1. Please check carefully.

P3 line 5 : I suppose that the authors refer to airborne measurements. There is a significant amount of ground based studies in the literature which already identified

and quantified SOA in the Mediterranean region.

P3 line 30. Specify the lower cut off diameter of the aircraft aerosol inlet.

P4 line 2-4. Please specify the time resolution of both SMPS and OPC

P5 line 14-22 : The PMF analysis requires much more details and explanation (Q/Qexp vs number of factors, residuals, boostrap/fpeak etc..). This could be added in the SI. Also explain why adding inorganic ions (SO4 and NO3) allows a better separation of the factors. Why NH4 ions are not considered?

I also suppose that the relative contributions of MOOA and LOOA reported in section 3.4.1 refer to the OA+SO4+NO3 mass concentration. Please clarify and discuss. Relative contributions to OA are from my point of view more relevant.

P5 line 24-31. Aerosol were apparently collected on TEM girds by means of a 2 stages impactor, but this information totally disappears in the discussion of the results. Is there any difference between the 2 particle size ranges? Also, what is the representativeness of the analyzed particles (at least 50 / gird) with respect to the whole particles population.

P6 section 3.1. Please add a map (with the flight tracks) in this section instead of in the SI. The authors state that one of the forest is a high isoprene emitter while the second is apparently more dominated by monoterpenes emissions. Considering the results presented in Tab1 and fig 1, isoprene and monoterpenes concentrations do not support this affirmation. Do you have any explanation?

P7 Section line 9-12. Even if the concentrations of aromatic VOC are low with respect to the total VOC concentrations measured here, they cannot be considered as negligible compared to isoprene and monoterpenes concentrations. They deserve more attention from the authors. For instance, what is B/T ratio. Does this ratio make sense with aged air masses? Are the concentrations of aromatics homogeneous all along the flight?

P7 line 13-20. I'm puzzled by the OH reactivity section. Either provide more details

(Atkinson and Arey 2003 is a 40 pages review, and no reference are provided for Waked et al), either this section can be removed from the paper.

P7 line 23-24. I suppose that the uncertainties provided here (and throughout the whole manuscript/tables/figures) correspond to the standard deviation associated to the average values. This must be specified clearly as well as which part of the flights have been averaged.

P7 line 29-31 : see main comment 4

P7 line 31-32 : "The high contribution of externally mixed particles indicates that most of aerosol particles were recently emitted from their source" sounds contradictory with the sentence p7 line 27 "The organic aerosol measured during all flights was well oxidised, . . ., with little evidence of fresh primary organic aerosol."

P8 line 3 : direct comparison between TEM analysis and AMS results are not relevant without a thoughtful analysis of representativeness of the particles analyzed by TEM with respect to the whole particles population.

P8 line 15. In order to support the assumption of the presence of organo-sulfates, did the authors check the ionic mass balance (SO4, NO3, NH4) from the AMS results? In other words, did they observe a deficit of NH4?

P8 line 30 : The authors should also mention and discuss the potential influence of industrial activities (Fos-Berre area) on the ultrafine mode observed during the easterly flights.

P9 line 6-7: 'with lowest values corresponding to highest fractions of fine particle concentrations (Fig. 5). These observations are in agreement with previous field studies' Considering fig 5 and the error bars, it's impossible to conclude.

P9 line 24-26. I don't understand why fig S8 is in the supporting information. From my point of view one of the main results from this study.

P9 line 30 : "MOOA, contributing 55%" of OA or OA+SO4+NO3? What are the O:C ratio for MOOA and LOOA?

P10 line 16 : "This m/z 91 was present in all OA mass spectra and was significantly higher for the LOOA than for the MOOA." Be more quantitative

P11 section 3.4.2. Not really convinced by the relevance of this section within the scope of this paper. Anyway. Why modelling results are only compared with measurements during the vertical profiles? How model and measurements compare during the low altitude legs?

P12 line 11-12. "The model estimates a significant contribution of isoprene SOA (approximately 15 to 35% to the total SOA). This cannot be confirmed by measurements due to the lack in a significant contribution of m/z 82 in the AMS spectra" This sentence makes no real sense. A very low contribution of m/z 82 means 1/ that there is a very little influence of isoprene SOA ; in that case the model is wrong or 2/ the isoprene SOA formation is formed through non-IEPOX pathways and in that case the question is how isoprene SOA is modelled?

Table 1 : please check the values and unit. Concentrations averaged during the whole flight or only low altitude legs? How the uncertainties are calculated (same for table 2)? Why C8 and C9 aromatics are not reported in the table?

Table 3 : What do you mean by "Pr"?

Figure 1 : Check values, add error bars, add % in the pie charts and add the flight codes (RF15 3006 etc..) in each panel in addition to a/, b/, .. (same for all figures)

Figure 5 : why no x error bars?

Figure 6 b : Why only 7 points represented here while ~100 are represented in fig S8?

Figure S6 : 11 RF represented, only 4 discuss in the manuscript. C8 aromatics are not only m-xylene.

---

## Referee Comment (RC2) · Anonymous Referee #1 · 16 Sep 2017

The authors present airborne measurements of particles and gases above Mediterranean forests during the ChArMEx campaign. Offline TEM analysis is also presented. These measurements are used to investigate the sources of SOA in this forest. PMF and the Polyphemus model was also used to estimate the contribution of various sources to SOA. This manuscript employs a nice combination of measurements and modeling to investigate SOA. However, I find numerous major and technical corrections that need to be addressed prior to publication in ACP, as listed below:

Major Comments:

P2 L7: You have defined "isoprene epoxydiols SOA" as "(IEPOX)", which is incorrect. Isoprene epoxydiols are compounds typically found in the gas phase and can be called IEPOX for short. They can go on to form isoprene epoxydiols-derived SOA, which is

often called IEPOX-SOA.

P3 L15-18: I recommend that some of this information about what took place during this campaign be moved to the methods section, and replaced with a general description of what you investigated in this particular manuscript.

P3 L22: Is ATR an acronym? Please define if possible. Also, for readers who are unfamiliar with this aircraft, please provide more general information in this paragraph. E.g., what is the general size of the aircraft, how large is the payload, did it sample anything other than meteorological parameters and aerosols?

P4 L5: Provide a reference for the cToF-AMS instrument. Also, assuming it was the Aerodyne instrument, it should be referred to as the "Aerodyne compact time-of-flight aerosol mass spectrometer (cToF-AMS)" and called cToF-AMS instead of C-ToF-AMS to be consistent with previous literature.

P4 L6: I think you mean "spatial" instead of "temporal"? What was your spatial resolution, i.e., how far does the plane travel in 40 s?

P4 L26: Was it a high resolution (HR) or unit mass resolution (UMR) PTR-MS?

P6 L13: Section 3.1 belongs as part of your Methodology section 2 above. This section describes where and when the measurements were taken (along with some information about general conditions of the atmosphere), but doesn't provide any science results.

P6 L13: When you introduce the four RF's in this section (and then use them throughout the paper), please change it to use a uniform style. In this version, you have everything from, e.g., "RF20" to "RF20 03/07" to "RF 03/07 RF20" to "RF0307" (in Fig. S1) all referring to the same flight. I recommend providing the date of each flight in the methodology and then only use "RF20" in the results section.

P6 L20: Wherever you decide to specify the dates of the flights, I strongly recommend that you use a date format such as "3 July" instead of "03/07" throughout the manuscript and figures. 03/07 could mean 7 March or 3 July depending on where in the world the

Interactive
comment

reader is.

P7 L11: When you say "...and aromatics)", do you mean C8- and C9-aromatics? This sentence doesn't make sense as is.

Table 1: All of the VOC concentrations are in pptV, not ppbV, correct? Please check the units. Also what does the +- value represent? Range? Standard deviation?

P7 L27: "...was well oxidized" is a subjective statement that doesn't add scientific value or understanding, please change.

P7 L28: "with little evidence of fresh primary organic aerosol": Please provide or cite your evidence for this statement. The O:C value of the bulk OA by itself does not provide information about what types of OA (primary vs secondary) constitute the bulk OA.

P8 L15: With the evidence that you've shown, I'm not really convinced that you can conclude that you're seeing organosulphates. You state that you don't have any visual evidence that there were other compounds e.g. ammonium sulphate present, but can you show evidence that NH4 (or N) was NOT present? That's what I would want to see to really back up your conclusion of organosulphates, particularly because you've included this conclusion in your abstract.

Figure 3: Please explain in the figure caption what each panel is showing. Also, in 3a), are all four panels showing the same particle? It looks to me like the substrate in the top left panel is at a different angle than it is in the other three panels (the substrate overlaps with the bottom left corner in all panels except the top left panel).

P8 L30: I do not see sufficient evidence to say that it is "likely" that the fine mode particles were linked to new particle formation. You haven't shown measurements of new particle formation or even particles <20nm in size (and it seems those measurements were not taken during these flights), nor have you discussed possible primary sources of particles such as cities, vehicular traffic, etc. Please change this text so that

all of your conclusions are backed up with evidence, or change the strength of your conclusions to reflect the uncertainties.

P9 L14: This section 3.4.1 and also 3.4.2 are really part of aerosol chemical properties. I think they belong much better as subsections to 3.3 (aerosol chemical properties) rather than 3.4 (aerosol physical properties). Alternatively, they can be standalone sections.

P9 L26: Yes, isoprene oxidation is likely leading to some SOA formation, but what about other precursors? Is there any correlation with, e.g., monoterpenes? I would like to see a more thorough analysis here that considers all measured SOA precursor gases.

P9 L31: In Fig. 6a, m/z 80 is shown as an organic peak (green). It should be red (sulphate), correct?

P10 L3: You state here and in other places that you are plotting as a function of photochemical age, which seems incorrect. Photochemical age would have units of time (hours or days), but you're plotting against the ratio of isoprene oxidation products to isoprene. It should be possible to convert this ratio to a unit of time, depending on whether the rate constants and all of the relevant reactions involving these compounds are known enough to do so. If not, I suggest that you say instead that you plot as a function of relative age, or simply as a function of the ratio which represents the relative age of the airmass.

P10 L15: What is the value of f91 in your two spectra, and how does this compare with the other cited studies that investigated the contribution of monoterpenes or other pathways to m/z 91? The f91 in your spectra appear to be very small, suggesting these pathways aren't important.

Table 3: Please fix the formatting, m/z 71 in the first column should not be bold. Also, what does Pr mean? Are these Pearson r correlation values? Or R^2? Please clarify.

P10 L34: "...we can conclude that the observed OA can probably be related to a non-IEPOX isoprene SOA.": This conclusion as written is not backed up by the evidence you are presenting. First, a given compound's contribution to photochemical activity (by which I think you mean OH reactivity, which is different) has little to do with it's ability to form SOA. Sure, isoprene may be the main contributor to OH reactivity (among the measured compounds), but what matters for SOA formation is the SOA yield, and you haven't discussed that in this work. Second, you are presenting some evidence that suggests that IEPOX-SOA was not present in substantial amounts, but then your only conclusion in this section of the manuscript should be that IEPOX-SOA was not an important contributor. You have presented no measure of non-IEPOX isoprene SOA, so you have no basis on which to speculate about the magnitude of the non-IEPOX isoprene SOA at this site, whether it is dominant, negligible, or somewhere between. I suggest you clarify that you find that IEPOX-SOA appears negligible at this site, and that non-IEPOX isoprene SOA may play a role but it's unclear how much.

P11 L4: It will be very difficult for the editor, reviewers, and future readers to properly interpret this modeling work if the full model details are not yet published in this work or elsewhere! It would seem improper for this manuscript to be published before the Chrit et al. manuscript (in which you say the details can be found) at least appeared in ACPD.

P12 L11: This final paragraph should be removed, it is repeated information and doesn't add to the manuscript. In its place, you could consider strengthening this section by adding a paragraph to compare the model results with your measurements that were presented earlier in the paper. Are your measurements consistent with the model? What new information have we learned? What are the next steps, e.g., what other model or measurement results would we need to learn more?

Technical Corrections:

P3 L23: Please change "preformed" to "performed".

P3 L29: Please change "chaneling" to "channeling".

P4 L2: Please change to "scanning mobility particle sizer".

P4 L17: Please specify explicitly which standard temperature and pressure, to eliminate any possible confusion.

P4 L26: Update the Waked et al. citation if possible.

P4 L26: Add "(VOC)" after the word "compounds" to define this acronym.

P4 L27: Add references for the PTR-MS instrument.

P5 L16: I believe the proper name to use here in the "PMF Evaluation Tool (PET)".

P5 L20: SQUIRREL is an acronym and should be capitalized.

P7 L18: Please change from Fig. A6 to Fig. S6.

Table 2: Specify the units, as well as what the +- values represent.

P8 L15: Change from A7 to S7. There are other locations in the manuscript with similar A instead of S, please correct all instances.

P9 L5: Please define what isopreneC and monoterpeneC are.

P9 L11: Change "rations" to "ratios".

P10 L7: m/z should be italicized in all places in the paper.

P11 L6: Your supplemental figures are not referenced in order. Your Fig. S3 was just referenced here for the first time. I'm not sure if ACP has strict guidelines about this, but it's common to number them in the order they appear in the manuscript.

Many figures: The text and labels in a lot of the figures are too small to read. I suggest you make the font larger for clarity.

---

## Author Comment (AC1) · 10 Jan 2018

**Authors present the results obtained during an airborne measurement campaigns over 2 Mediterranean forested areas (South of France). More precisely, they report results combining c-ToF-AMS, HS-PTR-MS, SP2, SMPS/OPC, Aerolaser (for formaldehyde measurements) as well as offline TEM analysis, from 4 flights (2 for each forest). Simulations performed with the Polyphemus model and comparisons with the measurements are also presented.**

**While the paper is well written with a clear structuration, significant work is still needed to make it suitable for publication in ACP. Important details are missing and some major issues can also be found here and there throughout the manuscript (despite 21 co-authors).**

The authors would like to thank the reviewers for their very constructive and informative comments. These comments and suggestions have helped us to improve the quality of our manuscript. Below we have responded to each of the reviewers comments. **The reviewers comments are in bold** and our reponses are in normal text.

**Main comments:**
**1- There are major inconsistencies between VOCs concentrations reported in the text (p2 line 6, for instance), in figure 1 and in table 1. Not only a question of units. Must be checked very carefully and corrected.**
P2, Line 6 has been corrected and the remainder of the text has been verified for inconsistencies.

**2- Regarding PMF analysis, much more details are needed (at least in the SI)**
Additional details are now included into the supplementary material, including figures S9 to S11 (Factor profiles, scaled residuals, Q/Qexp and fpeak analysis), as well as correlation with time series for 3 and 4 factor solutions.

**3- It's not clear if the average values reported in the text and tables refers to the whole flight or only to the low altitude legs. This should be clarified and homogenized throughout the text. Considering the vertical profiles presented in figure S4, whole flight averages make no real sense. These vertical profiles and their implications are not discussed in the text except for boundary layer heights (p6, line 20).**
The average values reported in the text represent only the low and constant altitude legs of the flight. This has been clarified in the text.

The vertical profiles are only used to determine the boundary layer heights, only data collected during the vertical profile measurements are used for the plots shown in Figure S3. The caption in Fig. S3 has been changed to clarify this.

Page 8, Line 2:
"In the following section we will report average values for different chemical species during low and constant altitude parts of the flights."

Figure S4. Vertical profiles of RH, formaldehyde, and isoprene + its oxidation products (MVK+MCR) for research flights: a) RF15 , b) RF20, c) RF21 , d) RF23 . Only data collected during the vertical profile measurement are used in these plots.

**4- Either the definition of externally mixed aerosol has recently changed either I don't understand the difference between fig 3a and b. This leads to important confusions in the discussion and conclusions.**
This paragraph has been re written to avoid confusion.

"As described in section 2.5, the chemical composition of aerosol particles collected on TEM grids was determined using EDS. At least 230 particles were analyzed during each flight providing information of particle size and composition. The absolute number of particles analyzed using offline electron microscopy is small in comparison to what is measured by online particle counters, however this technique provides us with a qualitative snap shot into particle mixing state, morphology and composition. Only filters from the submicron stages are discussed here and showed that at least 35 (± 5) % of all aerosol particles measured was made up of externally mixed amorphous (no evidence of a crystal structure) particles. EDS analysis of these amorphous particles were composed of homogeneously distributed C, O, and S (Fig. 4a i) ii) iii), the molecular structure of these particles is unknown. Externally mixed crystalline sulphate particles contributed 15 (± 5) % and 10 % were internally mixed amorphous C and crystalline sulphate (likely ammonium sulphate) species (Fig. 4b, Fig S4). The remaining fractions contained signals for sea-salt (Na Cl) and dust (Si, Ca) particles. "

**Other comments**
**R3.1 : P1 line 30-31. Direct comparisons between AMS and TEM analysis are not relevant. AMS refers to a mass concentration without distinction between internally and externally mixed aerosol particles and TEM analysis refers, at best, to a number concentration. Authors state that at least 50 particles per gird have been analysis. What is the representativeness of the analyzed particles with respect to the total particles population? (SMPS/OPC measurement can help to assess this representativeness)**

Yes, the authors agree that direct comparisons between AMS and TEM are not possible and therefore the text has been modified to avoid misleading the reader. In regard to the representativeness, total particle concentration observed during flights were in the range of 3289±1942 cm$^{-3}$, considering each sample was collected for approximately 15 minutes, this leads to about 14394 particles that could have been trapped by the grid. Given that TEM analysis of a substantial fraction of those particles is impossible, we have performed a sensitivity study of our findings in terms of particle population size by, increasing five-fold the TEM analysis and comparing previous results. Indeed, the new statistics compare quite well (OA contribution ranging between 55 and 75% (compared to previously obtained values ranging between 35 and 52%), S contribution ranging between 10 and 43% (compared previously with ranges of 11 and 25 %), indicating that already the number of particles analysed here led to robust statistics, showing that OA clearly dominants the aerosol spectra. The results throughout the manuscript have been updated to reflect the larger statistics.

**R3.2 P 1 line 32. "Externally mixed organic aerosols, were equally identified with S signals, which may suggest the presence of organo-sulphates" I really don't understand this sentence. (cf main**

**comment 4)**The text has been modified to avoid confusion. Please refer to the response to general comment 4.

**R3.3 P2 line 6 : "high mixing ratio of isoprene (2-4 ppbV) and oxidation products (0.6 and 1.2 ppbV)" These concentration ranges cannot be found in table 1 and figure 1. Please check carefully.**
These values have been corrected.

**R3.4 P3 line 5 : I suppose that the authors refer to airborne measurements. There is a significant amount of ground based studies in the literature which already identified and quantified SOA in the Mediterranean region.**
This phrase has been removed.

**R3.5 P3 line 30. Specify the lower cut off diameter of the aircraft aerosol inlet.**
The upper cutoff diameter of the aerosol inlet is specified, the lower cutoff diameter has not been measured. This inlet is both isokinetic and isoaxial and has a 50% sampling efficiency for aerosol particles with diameters of 4.5 µm (Crumeyrolle et al. 2013).

**R3.6 P4 line 2-4. Please specify the time resolution of both SMPS and OPC**
The text has been modified to include this information.
Aerosol particle number concentrations were measured using a scanning mobility particle sizer (SMPS) with 162 size channels for particle diameters ranging from 20 nm up to 400 nm, with a time resolution of 84 secs. Larger particles size distributions were measured using a GRIMM optical particle counter with 16 size channels from 265 nm up to 3 microns, with a time resolution of 1 second

**R3.7 P5 line 14-22 : The PMF analysis requires much more details and explanation (Q/Qexp vs number of factors, residuals, boostrap/fpeak etc..). This could be added in the SI.**
Additional details are now included into the supplementary material, including figures S9 to S11 (Factor profiles, scaled residuals, Q/Qexp and fpeak analysis), as well as correlation with timeseries for 3 and 4 factor solutions.

**R3.8 Also explain why adding inorganic ions (SO4 and NO3) allows a better separation of the factors. Why NH4 ions are not considered? I also suppose that the relative contributions of MOOA and LOOA reported in section 3.4.1 refer to the OA+SO4+NO3 mass concentration. Please clarify and discuss. Relative contributions to OA are from my point of view more relevant.**

Additional text has been included to explain this.
Inorganic species $SO_4$ and $NO_3$ followed similar temporal profiles to the organic species and are suspected as coming from similar processes (Pearson R OA SO4 = 0.70, OA NO3 = 0.84). By combining these m/z values into the PMF matrix it allowed us to perform an apportionment of the main aerosol sources and not just those related to the organic molecules. Combining the organic mass spectral fragments with those of nitrate and sulphate, provided us with additional mass concentrations to perform a more robust analysis on this aerosol data. We omitted NH4 from the analysis since the signal was noisy.

They were relative contributions to OA.

**R3.9 P5 line 24-31. Aerosol were apparently collected on TEM girds by means of a 2 stages impactor, but this information totally disappears in the discussion of the results. Is there any difference between the 2 particle size ranges? Also, what is the representativeness of the analyzed particles (at least 50 / gird) with respect to the whole particles population.**

The text has been updated and is shown above in response to general comment 4. Only particles from the submicron stage were analysed for this study to be coherent with the cut off diameters measured by the other instruments (SMPS, AMS). Additional TEM analyses have been performed so that we now have >230 particles per sample. However, still the fraction of particles collected on the electron microscopy grid is so small when compared with the total particle population measured by the SMPS/CPC/OPC. Nevertheless, this technique is providing us with a qualititative snap shot into particle mixing state morphology and composition and we believe the comparison should provide useful information to readers to understand the mixing states of measured particles. The text has been modified (see response to Response to general comment 4.).

"At least 230 particles were analysed for size and composition for filters collected during each flight. The absolute number of particles analysed using offline electron microscopy is negligible in comparison to what is measured by online particle counters, however this technique provides us with a qualititative snap shot into particle mixing state, morphology and composition."

**R3.10 P6 section 3.1. Please add a map (with the flight tracks) in this section instead of in the SI.**
This has been added. Figure S5 is now Figure 1

**R3.11 The authors state that one of the forest is a high isoprene emitter while the second is apparently more dominated by monoterpenes emissions. Considering the results presented in Tab1 and fig 1, isoprene and monoterpenes concentrations do not support this affirmation. Do you have any explanation?**

The nature of the vegetation in the two forested areas is very different with principally isoprene emitting trees in the OHP forest (Quercus pubescens) and during the western flights the vegetation is evergreen oaks (Quercus ilex) and Alpine pines (halepensis). However, the lifetime of isoprene in the atmosphere is considerably longer than for monoterpene species and results in a high atmospheric background. In addition to this the meteorology experienced on the different flights were different enough that the quantity of relative emitted concentrations of each VOC species were different, and makes it difficult to compare the absolute concentrations of one flight with the other.

**R3.12 P7 Section line 9-12. Even if the concentrations of aromatic VOC are low with respect to the total VOC concentrations measured here, they cannot be considered as negligible compared to isoprene and monoterpenes concentrations. They deserve more attention from the authors. For instance, what is B/T ratio. Does this ratio make sense with aged air masses? Are the concentrations of aromatics homogeneous all along the flight?**

A figure has been added to the supplementary material showing the temporal evolution of toluene and benzene as a function of time as well as a plot of total OA against the Toluene Benzene ratio. Additional text is included in the main part of the manuscript.

[Figure]

A time series plot of total organic aerosol (OA) with MCR+MVK+ISOPOOH shows a good relationship (Fig. S5a), and plotting the OA concentration against the ratio of (MCR+MVK+ISOPOOH)/isoprene provides us with a means to observe the evolution of the organic aerosol with the relative age of the air mass with respect to biogenic emissions. The ratios of (MCR+MVK+ISOPOOH)/isoprene measured during this flight are comparable to those measured over this forested area (0.4 to 0.8) (Zannoni et al., 2016). We observe a reasonable correlation (r = 0.46) and positive slope (b = 1.1) with increasing OA as the relative air mass age increases, suggesting that SOA formation may have originated from biogenic precursors (Fig. S5b). Similar plots were prepared using anthropogenic precursor gases benzene and toluene (Fig. S6) showing a negative correlation with increasing organic mass concentration of r = 0.35 and a slope of -0.56. These trends suggest that the increases in OA concentrations are primarily related to biogenic emissions.

**R3.13 P7 line 13-20. I'm puzzled by the OH reactivity section. Either provide more details (Atkinson and Arey 2003 is a 40 pages review, and no reference are provided for Waked et al), either this section can be removed from the paper.**

This section has been removed

**R3.14  P7 line 23-24. I suppose that the uncertainties provided here (and throughout the whole manuscript/tables/figures) correspond to the standard deviation associated to the average values. This must be specified clearly as well as which part of the flights have been averaged.**

Yes, these uncertainties correspond to standard deviations. This is now clearly specified throughout the manuscript.

**R3.15  P7 line 29-31 : see main comment 4**
EM imaging..
We have added more explanation, updated the figures and clarified the text in order to respond to the reviewers comments. Details of these changes are outlined above.

**R3.16  P7 line 31-32 : "The high contribution of externally mixed particles indicates that most of aerosol particles were recently emitted from their source" sounds contradictory with the sentence p7 line 27 "The organic aerosol measured during all flights was well oxidised, . . ., with little evidence of fresh primary organic aerosol."**
The text in this section has been updated. Changes are included above in response to previous comments.

**R3.17   P8 line 3 : direct comparison between TEM analysis and AMS results are not relevant without a thoughtful analysis of representativeness of the particles analyzed by TEM with respect to the whole particles population.**
We agree with the reviewer and have modified the text of this section accordingly. These modifications have been included above in response to other comments.

**R3.18 P8 line 15. In order to support the assumption of the presence of organo-sulfates, did the authors check the ionic mass balance (SO4, NO3, NH4) from the AMS results? In other words, did they observe a deficit of NH4?**
Yes, we did check but there was no significant deficit of NH4, the neutralization graph is included in the supplementary materials, and additional information has been included in the text. The plot is coloured by SO4 concentrations to show that sulphate concentrations were not associated with periods when the NH4pred/NH4meas ratio were low.

[Figure]

**Figure S11:** Average surface concentrations (in µg m$^{-3}$) of submicron organic matter ($OA_1$) simulated during the a) RF20 03/07. and b) RF23 07/07 flights. The flight path is shown with bold black lines.

Updated text P11, L7: "Aerosol concentrations measured by the cToFAMS appear to be fully neutralised with little evidence of acidity (Fig. S11), and the $NO_W$ concentrations measured during these flights varied from 6 up to 10 ppbV, however the average concentrations of NO are 0.30 ± 0.2 ppbV, suggesting that the real contributions of $NO_X$ (See section 2.3) is also likely to be low. There have been some reports of isoprene-derived SOA formation (hereafter isoprene SOA) in high-NO regions but the contribution of this pathway is considered to be much smaller (Jacobs et al., 2014).

The yield of formation of SOA from isoprene VOC precursor is relatively low compared to other biogenic species such as monoterpenes, and also compared with aromatics precursors (Ait-Halel et al., 2015). Given the lack of acidic aerosol and the measured NO concentrations (0.30 ppbV), we do not believe that isoprene derived SOA contributed significant amounts to the OA measured during these flights. Therefore, although isoprene concentrations are in abundance during these flights, it is unlikely to have contributed large amounts to the formation of the measured SOA. Given the increase in OA with the relative "biogenic" air mass age, we could suspect that other sources of SOA during these flights could be from terpene precursors. This is also coherent with the increase in the number concentrations of fine particles at lower IsopreneC/monoterpeneC ratios. It should also be noted that the yield of SOA formation from aromatic precursors is similar to that of monoterpenes, and that some contribution of anthropogenic SOA can be expected."

**R3.19  P8 line 30 : The authors should also mention and discuss the potential influence of industrial activities (Fos-Berre area) on the ultrafine mode observed during the easterly flights.**
The contribution of anthropogenic sources and specifically the Fos-Berre area are now included in the text

"This *m/z* 91 was present in all OA mass spectra and was significantly higher for the LOOA (f91 = 0.007) than for the MOOA (f91 = 0.003). The contribution of f91 to biogenic organic aerosol over coniferous forests (mainly monoterpenes emitters), varied from 0.005 up to 0.019, and has been shown to be due to the $C_7H_7^+$ fragment (Lee et al., 2016). Our contribution of f91 falls into background OA levels (Lee et al., 2016), showing that the contribution of monoterpene or other pathways to m/z 91 are not important. It should be noted that *m/z* 91 can also be associated with fragments of primary OA and the contribution of anthropogenic aerosols from the industrial  zone (Fos-sur-Mer) south of the flight area cannot be ruled out."

**R3.20  P9 line 6-7: 'with lowest values corresponding to highest fractions of fine particle concentrations (Fig. 5). These observations are in agreement with previous field studies'**

**Considering fig 5 and the error bars, it's impossible to conclude.**

The figure now uses the difference between a condensation particle counter (CPC) having a particle cut off diameter of 5 nm and the SMPS having a particle cut off of 17 nm, the difference in the total number measured by these two instruments represents nucleation mode particles in the range from 5 nm to 17 nm.  The error bars are still relatively high, however the overall trend in these measurements illustrate that the highest concentrations of fine particles are associated with lowest isoprene concentrations. The text has been changed accordingly

P9, Line 13: "Using the measurements available during these four flights, we investigated this relationship between biogenic VOC species and nucleation mode particles; we calculated the ratio of isopreneC/monoterpeneC (Carbon associated with Isoprene/Monoterpene) and compared it to the number concentration of nucleation mode particles. As a result of the low time resolution of the SMPS, we were limited to a small number of points per flight. Data was combined for all flights, giving average ratios of isopreneC/monoterpeneC varying between 0.05 and 8 (average 3 ± 1), with lowest values corresponding to highest fractions of fine particle concentrations (Fig. 6). Although the variation among points is high, the general trend of these observations is in agreement with previous field studies over mixed deciduous forests (Kanawade et al., 2011) and with laboratory studies in

controlled environments showing that high concentrations of monoterpenes, relative to isoprene, can favor new particle formation (Kiendler-Scharr et al., 2009)."

[Figure]

Figure 6. Difference between the CPC (cut off 5 nm) and the SMPS (cut off 17 nm) as a function of the IsopreneC/MonoterpeneC ratio. Values for the four biogenic flights are included, as well as average values over IsopreneC/MonoterpeneC ratios. Error barsrepresent ± 1 σ of the average CPC5nm – SMPS values. .The black line represents the linear correlation fit.

**R 3.21 P9 line 24-26. I don't understand why fig S8 is in the supporting information. From my point of view one of the main results from this study.**

This has been moved to the main part of the manuscript, Fig 7.

**R3.22  P9 line 30 : "MOOA, contributing 55%" of OA or OA+SO4+NO3? What are the O:C ratio for MOOA and LOOA?**
The percentages here refer to the contribution of each of the identified factors to the total resolved factors (which would include concentrations of OA+SO4+NO3).
The text has been updated to clarify this.

The O:C ratios are now included in the text.
The MOOA component is associated with inorganics species and had higher O:C ratios than the LOOA factor (0.83 and 0.76, respectively).
**
The two resolved factors include (i) a more oxidised organic aerosol (MOOA, contributing 55% to the resolved factors), containing high contributions from $m/z$ 44 and associated with inorganic peaks ($m/z$ 30, 46 (NO$_3$), and 48, 64, 80 (SO$_4$)) and (ii) a less oxidised organic aerosol species (LOOA, contributing 45%) with little contribution from inorganic $m/z$ (Fig. 6 a)).

**R3.23  P10 line 16 : "This m/z 91 was present in all OA mass spectra and was significantly higher for the LOOA than for the MOOA." Be more quantitative.**
The values of mz 91 for the two types of spectra are now included in the text, as well as comparison to prevous studies.
"This $m/z$ 91 was present in all OA mass spectra and was significantly higher for the LOOA (f91 = 0.007) than for the MOOA (f91 = 0.003). In recent work, on the characterisation of aerosol particles

over coniferous forests (mainly monoterpenes emitters) this has been shown to be due to the $C_7H_7^+$ fragment (Lee et al., 2016). It should be noted that *m/z* 91 can also be associated with fragments of primary OA and therefore other potential sources cannot be excluded."

**R3.24  P11 section 3.4.2. Not really convinced by the relevance of this section within the scope of this paper. Anyway. Why modelling results are only compared with measurements during the vertical profiles?  How model and measurements compare during the low altitude legs?**

Modeling results are compared to measurements during the whole flight.
We have added text to clarify this:
Line 22, p11: Modeling results are compared to measurements by averaging the measured and modelled concentrations during the whole flight, along the flight path.

**R3.25  P12 line 11-12. "The model estimates a significant contribution of isoprene SOA (approximately 15 to 35% to the total SOA). This cannot be confirmed by measurements due to the lack in a significant contribution of m/z 82 in the AMS spectra" This sentence makes no real sense.**

**A very low contribution of m/z 82 means**
**1/ that there is a very little influence of isoprene SOA ; in that case the model is wrong or**
**2/ the isoprene SOA formation is formed through non-IEPOX pathways and in that case the question is how isoprene SOA is modelled?**

The modelling of SOA formation is based on chamber experiments. The formation pathway used to represent the observed SOA in chamber experiments in low NOx environments is a non-IEPOX pathway, and the isoprene SOA surrogate is assumed to be a methylmethyl dihydroxy dihydroperoxide, as specified line 13 page 11.

For clarity on how the model was made, line 12 page 11, we have added text. The modelling of SOA formation is based on smog chamber experiments, which provide information on SOA yield as a function of organic mass concentration for each precursor, and uses an Odum approach (Odum et al., 1996). Stoechiometric coefficients of SOA surrogates and their saturation vapor pressures are selected to fit data from smog chambers. Candidates for SOA surrogates are estimated from the literature (Couvidat et al., 2012).

**R3.26 Table 1 : please check the values and unit. Concentrations averaged during the whole flight or only low altitude legs? How the uncertainties are calculated (same for table 2)? Why C8 and C9 aromatics are not reported in the table?**
The units and values are corrected in Table 1. The concentrations are averaged during low and constant altitude parts of the flight. The uncertainties are calculated from the standard deviations of the averaged values. The C8:C9 aromatics (combined) are now included in table 1

The table caption has been updated to clarify this.
Table 1. Mean concentrations of the different gas phase species measured during low and constant altitude parts of each flight. The error represents ± 1σ on all the measurements.
**R3.27 Table 3 : What do you mean by "Pr"?**
Pr reprents the Pearsons r correlation coefficient. The Table caption has been changed to include the necessary information.

**R3.28 Figure 1 : Check values, add error bars, add % in the pie charts and add the flight codes (RF15 3006 etc..) in each panel in addition to a/, b/, (same for all figures)**

These figures have been updated as requested by the reviewer.

Figure 2. Contribution of the different measured gas phase species aboard the ATR-42: a) RF15, b) RF20, c) RF21, d) RF23. The pie charts illustrated in each figure represent the contribution of all VOC species *except those of methanol and acetone*.

[Figure]

**R3.29 Figure 5 : why no x error bars?**
The Y error bars are calculated by averaging the number concentration as a function of fixed size bins for the MCR+MVK/ISOPRENE. The error bars are standard deviations based on the mean variation of measurements within the fixed bins of the MCR+MVK/Isoprene values.

**R3.30 Figure 6 b : Why only 7 points represented here while _100 are represented in fig S8?**
The points were averaged over bins of 0.1 (MCR+MVK+ISOPOOH/ISOPRENE). This allowed us to have a clearer representation of the trends in the data. This is clarified in the figure legend.

**R3.31 Figure S6 : 11 RF represented, only 4 discuss in the manuscript. C8 aromatics are not only m-xylene.**

This figure is now changed to only include the 4 flights in the manuscript rather than the whole CHARMEX campaign. The legend is changed to include C8, C9 aromatics and not only m-xylene.

---

## Author Comment (AC2) · 10 Jan 2018

**The authors present airborne measurements of particles and gases above Mediterranean forests during the ChArMEx campaign. Offline TEM analysis is also presented. These measurements are used to investigate the sources of SOA in this forest. PMF and the Polyphemus model was also used to estimate the contribution of various sources to SOA. This manuscript employs a nice combination of measurements and modeling to investigate SOA. However, I find numerous major and technical corrections that need to be addressed prior to publication in ACP, as listed below:**

The authors would like to thank the reviewers for their very constructive and informative comments. These comments and suggestions have helped us to improve the quality of our manuscript. Below we have responded to each of the reviewers comments. **The reviewers comments are in bold** and our reponses are in normal text.

**Major Comments:**
**R1.1 P2 L7: You have defined "isoprene epoxydiols SOA" as "(IEPOX)", which is incorrect. Isoprene epoxydiols are compounds typically found in the gas phase and can be called IEPOX for short. They can go on to form isoprene epoxydiols-derived SOA, which is often called IEPOX-SOA.**
Thank you for this correction. We have replaced IEPOX with IEPOX-SOA throughout the manuscript.

**R1.2 P3 L15-18: I recommend that some of this information about what took place during this campaign be moved to the methods section, and replaced with a general description of what you investigated in this particular manuscript.**
The text has been updated:

"In this work we present observations from four research flight over the forested Mediterranean region. The objectives of these flights were to characterize aerosol chemical and physical properties and investigate the origin of SOA over these forested areas.

**R1.3 P3 L22: Is ATR an acronym? Please define if possible. Also, for readers who are unfamiliar with this aircraft, please provide more general information in this paragraph. E.g., what is the general size of the aircraft, how large is the payload, did it sample anything other than meteorological parameters and aerosols?**
The accroynm ATR represents Avions de Transport regional. I have included additional information on the aircraft.

"The ATR (Avion de Transport Régional) is a turbo propeller aircraft of approximately 23 m long and 25 m wide, having a payload of about 4.6 ton (www.atraircraft.com)."

We moved the following sentence describing how aerosol measurements are made into section 2.2

"In order to sample aerosol particle species, a forward facing aerosol inlet was fitted in place of a side window. This inlet is designed with an outer sleeve for channeling air and a large tube radius with low curvature to limit particle losses due to deposition. This inlet is both isokinetic and isoaxial and has a 50% sampling efficiency for aerosol particles with diameters of 4.5 μm (Crumeyrolle et al. 2013). From the aerosol inlet the sampled aerosols are directed through a manifold to a number of different instruments"

The technical details of how gas-phase measurements are made are included in section 2.3.

**R1.4 P4 L5: Provide a reference for the cToF-AMS instrument. Also, assuming it was the Aerodyne instrument, it should be referred to as the "Aerodyne compact time-of-flight aerosol mass spectrometer (cToF-AMS)" and called cToF-AMS instead of C-ToF-AMS to be consistent with previous literature.**
We have included a citation to Drewnick et al., 2005, and changed C-ToF-AMS to cToFAMS.

Drewnick, F., S.S. Hings, P.F. DeCarlo, J.T. Jayne, M. Gonin, K. Fuhrer, S. Weimer, J.L. Jimenez, K.L. Demerjian, S. Borrmann, D.R. Worsnop. A new Time-of-Flight Aerosol Mass Spectrometer (ToF-AMS): Instrument Description and First Field Deployment, *Aerosol Science and Technology*, 39:637-658, 2005. PDF (July 2005)

**R1.5 P4 L6: I think you mean "spatial" instead of "temporal"? What was your spatial resolution, i.e., how far does the plane travel in 40 s?**
We changed temporal to spatial and added text

"(aircraft covers approximately 5 km in 40 sec)".

**R1.6 P4 L26: Was it a high resolution (HR) or unit mass resolution (UMR) PTR-MS?**
The PTR-MS is a unit mass resolution instrument. We have added this information.
"For measurements of volatile organic compounds (VOC), a unit mass resolution  proton-transfer-reaction mass spectrometer (PTR-MS)…."

**R1.7 P6 L13: Section 3.1 belongs as part of your Methodology section 2 above. This section describes where and when the measurements were taken (along with some information about general conditions of the atmosphere), but doesn't provide any science results.**
This section has now been moved to section 2.6, and is highlighted in green.

**R1.8 P6 L13: When you introduce the four RF's in this section (and then use them throughout the paper), please change it to use a uniform style. In this version, you have everything from, e.g., "RF20" to "RF20 03/07" to "RF 03/07 RF20" to "RF0307" (in Fig.S1) all referring to the same flight. I recommend providing the date of each flight in the methodology and then only use "RF20" in the results section.**
Flight names have been changed throughout the manuscript to RF"flight number".

**R1.9 P6 L20: Wherever you decide to specify the dates of the flights, I strongly recommend that you use a date format such as "3 July" instead of "03/07" throughout the manuscript and figures. 03/07 could mean 7 March or 3 July depending on where in the world the reader is.**
This has been specified clearly in the tables and elsewhere we use only the flight number.

**R1.10 P7 L11: When you say ". . .and aromatics)", do you mean C8- and C9-aromatics? This sentence doesn't make sense as is.**

Yes, the text and tables have been updated to include this.

As written in the manuscript, Page 7, Line 15: "The principal VOC species measured by the PTRMS during all flights were acetone (*m/z* 59) and methanol (*m/z* 33), followed by isoprene (*m/z* 69) and its oxidation products (MVK + MCR + ISOPOOH) (*m/z* 71), and then VOC species representative of monoterpenes emissions (*m/z* 137). Anthropogenic VOC species (*m/z* 93 (toluene), *m/z* 79 (benzene), and C8- and C9 aromatics) never contributed more than 5% to the total VOC measured (Table 1)."

**R1.11 Table 1: All of the VOC concentrations are in pptV, not ppbV, correct? Please check the units. Also what does the +- value represent? Range? Standard deviation?**
The table has been updated to include the units. The +- values represents the standard deviation on the measurements during the flights. The table caption has been updated to make this clear.

Table 1. Mean concentrations of the different gas phase species measured aboard each flight. The error represents $\pm 1\sigma$ on all the measurements.

**R1.12 P7 L27: ". . .was well oxidized" is a subjective statement that doesn't add scientific value or understanding, please change.**
**R1.13 P7 L28: "with little evidence of fresh primary organic aerosol": Please provide or cite your evidence for this statement. The O:C value of the bulk OA by itself does not provide information about what types of OA (primary vs secondary) constitute the bulk OA.**

The sentence has been modified and additional text has been added.

P8, L8:The organic aerosol measured during all flights had high O:C ratios of 1.05 ($\pm$ 0.05), high f44 > 0.2 and corresponding low f43 < 0.6, suggesting that the majority of the organic aerosol was secondary, with little influence from fresh primary organic aerosol.

**R1.14 P8 L15: With the evidence that you've shown, I'm not really convinced that you can conclude that you're seeing organosulphates. You state that you don't have any visual evidence that there were other compounds e.g. ammonium sulphate present, but can you show evidence that NH4 (or N) was NOT present? That's what I would want to see to really back up your conclusion of organosulphates, particularly because you've included this conclusion in your abstract.**

The particle does include N. However, we do not believe that this N is related to ammonium sulfate because ammonium sulfate particles can generally be easily distinguished using TEM as they have a well-defined crystal structure and are beam sensitive as seen in the particle Figure b. However, from TEM analysis we cannot be certain that these particles are organosulfates. Changes to the figure and the text are shown below.

**a) Externally mixed organic aerosol particle**

[Figure]

**b) Internally mixed organic aerosol particle with sulfate**

[Figure]

Figure 4. a) i)Example of an amorphous particle deposited on a carbon substrate. EDS mapping analysis showing signals for ii) C Carbon, iii) O Oxygen and iv) S Sulfur.; b) Internally mixed amorphous particles with signals, i) before and ii) after electron beam damage. EDS analysis showing signals for iii) C carbon, iv) O Oxygen, and v) S Sulfur.

"As described in section 2.5, the chemical composition of aerosol particles collected on TEM grids was determined using EDS. At least 230 particles were analyzed during each flight providing information of particle size and composition. The absolute number of particles analyzed using offline electron microscopy is small in comparison to what is measured by online particle counters, however this technique provides us with a qualitative snap shot into particle mixing state, morphology and composition. Only filters from the submicron stages are discussed here and showed that at least 35 (± 5) % of all aerosol particles measured was made up of homogenously mixed amorphous (no evidence of a crystal structure) particles. EDS analysis of these amorphous particles were composed of homogeneously distributed C, O, and S (Fig. 4a i) ii) iii). Externally mixed crystalline sulphate particles contributed 15 (± 5) % (likely ammonium sulphate) and 10 % were internally mixed amorphous C and crystalline sulphate (likely ammonium sulphate) species (Fig. 4b, Fig S4). The remaining fractions contained signals for sea-salt (Na Cl) and dust (Si, Ca) particles. "

**R1.15 Figure 3: Please explain in the figure caption what each panel is showing. Also, in 3a), are all four panels showing the same particle? It looks to me like the substrate in the top left panel is at a different angle than it is in the other three panels (the substrate overlaps with the bottom left corner in all panels except the top left panel).**
Yes, they are the same particle. In the microscopic analysis, two different imaging systems are used in the TEM and scanning TEM (STEM) for the EDS mapping. Switching between these two systems results in the images looking slightly rotated.

The figure and figure caption has been updated.

"Figure 3. a) i) Example of an amorphous particle deposited on a carbon substrate. EDS mapping analysis showing signals for ii) C Carbon, iii) O Oxygen and iv) S Sulfur.; b) Internally mixed

amorphous particles with signals, i) before and ii) after electron beam damage. EDS analysis showing signals for iii) C carbon, iv) O Oxygen, and v) S Sulfur."

**R1.16 P8 L30: I do not see sufficient evidence to say that it is "likely" that the fine mode particles were linked to new particle formation. You haven't shown measurements of new particle formation or even particles <20nm in size (and it seems those measurements were not taken during these flights), nor have you discussed possible primary sources of particles such as cities, vehicular traffic, etc. Please change this text so that all of your conclusions are backed up with evidence, or change the strength of your conclusions to reflect the uncertainties.**

We have changed this analysis to include particle concentrations measured using the condensation particle counters (CPC). We calculated the difference of the total number concentration measured by the CPC (cut-off diameter 5 nm) with the total particle concentration measured by the SMPS (starting diameter of 17 nm), giving us the total number concentration of the nucleation mode particles. The figure is now showing a plot of the nucleation mode particles against the ratio of Isoprene C and monoterpene C. This plot is showing that there is a higher fraction of nucleation mode particles measured when higher contributions of monoterpenes (relative to isoprene) are present.

[Figure]

Figure 6. Ratios of IsopreneC/MonoterpeneC plotted as a function of the nucleation mode particles (difference between the CPC (cut off 5 nm) and the SMPS (cut off 17 nm)). Values for the four biogenic flights are included, as well as average values (black contained circles) calculated over a number of IsopreneC/MonoterpeneC ratios (size bins of 0.5), Error bars represent $\pm$ 1 $\sigma$ of the average CPC5nm – SMPS values.

We have included throughout the manuscript discussion of the contribution of anthropogenic species. These are listed below.

P7, Line 19: Despite not contributing large quantities to the total VOC measured we cannot ignore the presence of the anthropogenic VOC species measured during all flights. We used the ratios of toluene to benzene to assess the contribution of anthropogenic emissions. The highest anthropogenic contributions were measured during RF15, when the ratio of toluene/benzene was 1.54. Values greater than 2 are generally found close to urban sources (Ait-Helal et al., 2014). Other evidence of

anthropogenic influence is the low enhancement ratios of HCHO (Formaldehydel)/CH$_3$CHO (Acetaldehyde), for RF15, this ratio was calculated to be +0.56, whereas for the other flights over forested regions, these ratios were calculated to be 4.8 (RF20) and 3.9 (RF23). For RF15, air masses arrived from the north of France (Fig. S2), likely bringing some anthropogenic influence from the mainland. The easterly flights were principally influenced by local or southerly airmasses, were possibly influenced by emissions from Marseille or from the Fos Berre industrial area. Full details of the VOC measurements aboard the aircraft will be provided in Waked et al. (in prep).

P10, Line 25: Plotting these two factors as a function of air mass age using anthropogenic VOC species (ratio of benzene/toluene), we observe a relatively flat and decreasing trend. These observations would suggest the contribution of anthropogenic precursors, although not insignificant, play a lesser role in the formation of the SOA measured during these flights.

P13, Line 24: The model results estimates overall contribution of 66% biogenic species and approximately 30% aromatic. This work provides a unique insight into the formation of SOA far from urban sources in biogenic dominated regions. It highlights how even at background forested sites throughout Europe, the impact of urban emissions of SOA formation is not negligible.

**R1.17 P9 L14: This section 3.4.1 and also 3.4.2 are really part of aerosol chemical properties. I think they belong much better as subsections to 3.3 (aerosol chemical properties) rather than 3.4 (aerosol physical properties). Alternatively, they can be standalone sections.**
The layout of the paper has been changed so that these sections are now stand alone sections

**R1.18 P9 L26: Yes, isoprene oxidation is likely leading to some SOA formation, but what about other precursors? Is there any correlation with, e.g., monoterpenes? I would like to see a more thorough analysis here that considers all measured SOA precursor gases.**

The contribution of monoterpene emissions is very weak in all of these flights, but the yield for the formation of SOA from monoterpene emissions is much higher.

Additional discussions are included throughout the manuscript. Please see response to question R1.16

**R1.19 P9 L31: In Fig. 6a, m/z 80 is shown as an organic peak (green). It should be red (sulphate), correct?**
This figure has been updated.

[Figure]

Figure 6. a) A two factor solution determined from PMF analysis of the biogenic research flights. a) i) The more oxidised organic aerosol (MOOA) associated with inorganic peaks for sulphate (red) and nitrate (blue) ,a)ii) the less oxidised organic aerosol (LOOA) with a lower contribution of inorganic peaks. b) Variations of these two species with aging airmass (using MCR+MVK+ISOPOOH over isoprene as a proxy for photochemical age of airmass.

**R1.20 P10 L3: You state here and in other places that you are plotting as a function of photochemical age, which seems incorrect. Photochemical age would have units of time (hours or days), but you're plotting against the ratio of isoprene oxidation products to isoprene. It should be possible to convert this ratio to a unit of time, depending on whether the rate constants and all of the relevant reactions involving these compounds are known enough to do so. If not, I suggest that you say instead that you plot as a function of relative age, or simply as a function of the ratio which represents the relative age of the airmass.**

We have changed the text throughout this discussion to make this clear.

Page 10, Line 1: "A time series plot of total organic aerosol (OA) with MCR+MVK+ISOPOOH shows a good relationship (Fig. S5a), and plotting the OA concentration against the ratio of (MCR+MVK+ISOPOOH)/isoprene provides us with a means to observe the evolution of the organic aerosol with the relative age of the air mass with respect to biogenic emissions. The ratios of (MCR+MVK+ISOPOOH)/isoprene measured during this flight are comparable to those measured over this forested area (0.4 to 0.8) (Zannoni et al., 2016)."

Page 10, Line 25: "Plotting these two species as a function of the relative air mass age we observe a significant increase of the LOOA species with air mass age until a maximum is reached at ratios of 0.65. MOOA remains relatively stable, indicating an independent source. Plotting these two factors as a function of a proxy of air mass age using anthropogenic VOC species (ratio of benzene/toluene), we

observe a relatively flat and decreasing trend. These observations would suggest the contribution of anthropogenic precursors although not insignificant; play a lesser role in the formation of the SOA measured during these flights."

Page 11, Line 13: "The yield of formation of SOA from isoprene VOC precursor is relatively low compared to other biogenic species such as monoterpenes, and also compared with aromatics precursors (Ait-Helal et al., 2014). Given the lack of acidic aerosol and the measured NO concentrations (0.30 ppbV), we do not believe that isoprene-derived SOA contributed significant amounts to the OA measured during these flights. Therefore, although isoprene concentrations are in abundance during these flights, it is unlikely to have contributed large amounts to the formation of the measured SOA. Given the increase in OA with the relative "biogenic" air mass age, we could suspect that other sources of SOA during these flights could be from terpene precursors. This is also coherent with the increase in the number concentrations of fine particles at lower IsopreneC/monoterpeneC ratios. It should also be noted that the yield of SOA formation from aromatic precursors is similar to that of monoterpenes, and that some contribution of anthropogenic SOA can be expected."

**R1.21 P10 L15: What is the value of f91 in your two spectra, and how does this compare with the other cited studies that investigated the contribution of monoterpenes or other pathways to m/z 91? The f91 in your spectra appear to be very small, suggesting these pathways aren't important.**

The discussion in this section has been changed and this phrase has been removed.
*"This m/z 91 was present in all OA mass spectra and was significantly higher for the LOOA than for the MOOA. In recent work, on the characterisation of aerosol particles over coniferous forests (mainly monoterpenes emitters) this has been shown to be due to the $C_7H_7^+$ fragment (Lee et al., 2016). It should be noted that m/z 91 can also be associated with fragments of primary OA and therefore other potential sources cannot be excluded."*

**R1.22 Table 3: Please fix the formatting, m/z 71 in the first column should not be bold. Also, what does Pr mean? Are these Pearson r correlation values? Or R^2? Please clarify.**
The table formating is corrected. Pr represents the Pearson r correlation values. This is now clearly clarified in the table caption.
"Table 3: Pearson r (Pr) correlations for different time series during RF 20 and RF 23."

**R1.23 P10 L34: ". . .we can conclude that the observed OA can probably be related to a non-IEPOX isoprene SOA.":**

**This conclusion as written is not backed up by the evidence you are presenting;**

**First, a given compound's contribution to photochemical activity (by which I think you mean OH reactivity, which is different) has little to do with it's ability to form SOA. Sure, isoprene may be the main contributor to OH reactivity (among the measured compounds), but what matters for SOA formation is the SOA yield, and you haven't discussed that in this work.**

We have removed the paragraph discussing OH reactivity and have included additional text of the SOA yield associated with different VOC precursors. The formation of SOA from different precursor species is also discussed in the modelling section

Based on the reviewers comments we have modified the text in this section. These changes are included above in response to R1.20 "

**Second, you are presenting some evidence that suggests that IEPOX-SOA was not present in substantial amounts, but then your only conclusion in this section of the manuscript should be that IEPOX-SOA was not an important contributor. You have presented no measure of non-IEPOX isoprene SOA, so you have no basis on which to speculate about the magnitude of the non-IEPOX isoprene SOA at this site, whether it is dominant, negligible, or somewhere between.**

**I suggest you clarify that you find that IEPOX-SOA appears negligible at this site, and that non-IEPOX isoprene SOA may play a role but it's unclear how much.**

Thank you for your comments and suggestions. The text has been modified following your suggestion.

The text has been updated.
Page 11, Line 13 "The yield of formation of SOA from isoprene VOC precursor is relatively low compared to other biogenic species such as monoterpenes, and also compared with aromatics precursors (Ait-Helal et al., 2014). Given the lack of acidic aerosol and the measured NO concentrations (0.30 ppbV), we do not believe that isoprene derived SOA contributed significant amounts to the OA measured during these flights. Therefore, although isoprene concentrations are in abundance during these flights, it is unlikely to have contributed large amounts to the formation of the measured SOA. Given the increase in OA with the relative "biogenic" air mass age, we could suspect that other sources of SOA during these flights could be from terpene precursors. This is also coherent with the increase in the number concentrations of fine particles at lower IsopreneC/monoterpeneC ratios. It should also be noted that the yield of SOA formation from aromatic precursors is similar to that of monoterpenes, and that some contribution of anthropogenic SOA can be expected."

**R1.24 P11 L4: It will be very difficult for the editor, reviewers, and future readers to properly interpret this modeling work if the full model details are not yet published in this work or elsewhere! It would seem improper for this manuscript to be published before the Chrit et al. manuscript (in which you say the details can be found) at least appeared in ACPD.**
This manuscript is now published in ACPD, in the same CHARMEX special issue.
Chrit, M., Sartelet, K., Sciare, J., Pey, J., Marchand, N., Couvidat, F., Sellegri, K., and Beekmann, M.: Modelling organic aerosol concentrations and properties during ChArMEx summer campaigns of 2012 and 2013 in the western Mediterranean region, Atmos. Chem. Phys., 17, 12509-12531, https://doi.org/10.5194/acp-17-12509-2017, 2017.

**R1.25 P12 L11: This final paragraph should be removed; it is repeated information and doesn't add to the manuscript. In its place, you could consider strengthening this section by adding a paragraph to compare the model results with your measurements that were presented earlier in the paper. Are your measurements consistent with the model? What new information have we**

**learned? What are the next steps, e.g., what other model or measurement results would we need to learn more?**

This has been removed and the discussion on what are the next steps etc are included in the conclusion.

Technical Corrections:

**R1.26 P3 L23: Please change "preformed" to "performed".**
This has been changed.

**R1.27 P3 L29: Please change "chaneling" to "channeling".**
This has been changed.

**R1.28 P4 L2: Please change to "scanning mobility particle sizer".**
This has been changed.

**R1.29 P4 L17: Please specify explicitly which standard temperature and pressure, to eliminate any possible confusion.**
This information has been added.
All reported concentrations are in standard temperature and pressure (used here 22$^{\circ}$C, 950 hPa).

**R1.30 P4 L26: Update the Waked et al. citation if possible.**
This paper is still in preparation

**R1.31 P4 L26: Add "(VOC)" after the word "compounds" to define this acronym.**
This has been changed.

**R1.32 P4 L27: Add references for the PTR-MS instrument.**
A reference is already included in the text.
Full details of the PTR-MS configuration on-board and operating conditions are provided in Borbon et al. (2013)

**R1.34 P5 L16: I believe the proper name to use here in the "PMF Evaluation Tool (PET)".**
This has been changed.

**P5 L20: SQUIRREL is an acronym and should be capitalized.**
This has been changed.

**P7 L18: Please change from Fig. A6 to Fig. S6.**
This has been changed.

**Table 2: Specify the units, as well as what the +- values represent.**
Units have been added to the table and the caption has been updated.
Table 2. Concentrations ($\mu$g m$^{-3}$) of the different chemical species measured aboard each flight during level low altitude legs., error values are standard deviations calculated on the mean values of the measurements.

**P8 L15: Change from A7 to S7. There are other locations in the manuscript with similar**
A instead of S, please correct all instances.
This has been changed, and the manuscript has been searched for similar corrections.

**P9 L5: Please define what isopreneC and monoterpeneC are.**
Additional information has been included
isopreneC/monoterpeneC (Carbon associated with Isoprene/Monoterpene)

**P9 L11: Change "rations" to "ratios".**
This has been changed.

**P10 L7: m/z should be italicized in all places in the paper.**
This has been changed.

**P11 L6: Your supplemental figures are not referenced in order. Your Fig. S3 was just referenced here for the first time. I'm not sure if ACP has strict guidelines about this, but it's common to number them in the order they appear in the manuscript.**
The supplementary figures have been reorganized to be consistent with the text.

**Many figures: The text and labels in a lot of the figures are too small to read. I suggest you make the font larger for clarity.**
The figures have been updated.

---

## Editor Comment (EC1) · X. Querol (Editor) · 11 Jan 2018

Dear author and referees #1 and 2,

Thank you very much to the referee's comments that certainly helped to improve a lot the paper. Thank you also to the main author (and other authors) of this paper that made substantial changes following these comments and that certainly will improve the presentation of the results. Errors are natural in human activities, including writing paper on a very complex and interesting ones, as it is the case), but I agree with the comments raised by a referee on the too large number of them found by both reviewers in the originally submitted paper, taking into account that it has 21 co-authors. My questions are, have all authors read the paper that they are signing, did all contribute

to elaborate or revise the submitted article? Authorship rules in nowadays scientific articles should be revised in my opinion. In any case the revision procedure helped to greatly improve the presentation of interesting scientific results, thanks to the work of authors (probably not all of them) and the referees' support. Thanks to all of them. Please, send us the revised version a.s.a.p. to continue the revision process. Yours sincerely,

Xavier Querol
* * *

---

## Author Response (AR2)

**The authors have satisfactorily responded to all my comments and questions and significantly modified the manuscript accordingly. However, I still have some major issues regarding the revised text.**

We would like to thank the reviewer for taking the time to review the updated version of this manuscript. Our response to each of the comments and suggests are outlined below. **The reviewers comments are in bold and shaded grey**. The authors response is in plain black (reply to comments/original text) or red (updated) text.

**1/ P7, lines 21-27.** **The toluene to benzene ratio cannot be used to "assess contribution of anthropogenic emissions". This ratio only provides information regarding the photochemical age (or the integrated OH exposure) of the anthropogenic fraction of the whole VOC mixture. A T/B ratio of 1.54 do not indicate a significant anthropogenic contribution, but an integrated OH exposure of the anthropogenic inputs of about 16.106 molecules.cm$^3$.h (i.e. a photochemical age of about 11h, considering an initial (T/B) of 2, which is a little bit low from my opinion, and an OH concentration of 1.5 106 molecules cm$^{-3}$) and this whatever is the contribution of anthropogenic VOCs to the whole VOC/SVOC mixture.**

**In this study and considering the data shown in fig S6, benzene and toluene signals are very noisy (because very close to the dl of the instrument) which probably imply an important variability of the T/B ratio not directly related to the photochemical age of the potential anthropogenic inputs.**

**Also, with such low toluene concentrations and although monoterpenes concentrations are low, the interference from the monoterpenes fragmentation's in the drift tube (m/z93 representing classically about 5-10% of m/z137, depending on the E/N) could potentially be significant for the T/B ratio. While it's always interesting to have a look to this ratio, I'd put much less emphasis on it and, above all, I'd used it wisely.**

The text has been updated accordingly

Despite not contributing large quantities to the total VOC measured, we cannot ignore the presence of the anthropogenic VOC species measured during all flights (Table 1 and Fig.2). During westerly flights (RF15 and RF21), air masses arrived from the north (Fig. S2), possibly transporting accumulated anthropogenic emissions from over mainland France. Easterly flights (RF20 and RF23) being principally influenced by local or southerly air masses, are likely impacted by anthropogenic activities over the Marseille and Fos Berre industrial area.

**2/ P7, lines 27-30. An enrichment factor of a parameter X is classically defined as follows: ER=X(t)/X(t0). In the case of the ratio HCHO/CH3CHO what is the reference (t0)? Authors must also specify how HCHO has been measured in the experimental section. If measured by PTRMS, are the HCHO concentrations corrected by the RH? Although a throughout analysis of the VOCs is expected in a future paper, this section must be clarified to make it understandable and meaningful.**

**To summarize, the whole section 3.1 needs to be significantly improved and developed before publication.**

Original text "Other evidence of anthropogenic influence is the low enhancement ratios of HCHO (Formaldehyde)/CH₃CHO (Acetaldehyde), for RF15, this ratio was calculated to be +0.56, whereas for the other flights over forested regions, these ratios were calculated to be 4.8 (RF20) and 3.9 (RF23). Full details of the VOC measurements aboard the aircraft will be provided in Waked et al. (in prep)."

EF: We removed this sentence. The section has been updated as follows.

The principal VOC species measured by the PTR-MS during all flights were acetone (*m/z* 59) and methanol (*m/z* 33), followed by isoprene (*m/z* 69) and its oxidation products (MVK + MACR + ISOPOOH) (*m/z* 71) and then VOC species representative of monoterpenes emissions (*m/z* 137). Isoprene and its oxidation products showed a high temporal variation during flights suggesting a more local influence of these VOC species. Monoterpene VOCs, having a short atmospheric lifetime were measured in low concentration with little temporal evolution. Anthropogenic VOC species (*m/z* 93 (toluene), *m/z* 79 (benzene), and C8- and C9 aromatics) never contributed more than 5% to the total VOC measured (Table 1, Fig. 2). Despite this, we cannot ignore the presence of the anthropogenic VOC species measured during all flights. During westerly flights (RF15 and RF21), air masses arrived from the north (Fig. S2), possibly transporting accumulated anthropogenic emissions from over mainland France. Easterly flights (RF20 and RF23) being principally influenced by local or southerly air masses, are likely impacted by anthropogenic activities over the Marseille and Fos Berre industrial area.

**3/P8, line 11-21. This is now much clearer. I'd change the title of fig 4a accordingly: "homogeneously mixed amorphous organic aerosol particle"**

These changes have been made.

a) Homogeneously mixed amorphous organic aerosol particle

b) Internally mixed organic aerosol particle with sulfate

[Figure]

**4/P10 lines 6-9. I agree with this conclusion, but the caveats developed above must be considered here.**

The text has been updated: We observe a reasonable correlation (r = 0.46) and positive slope (b = 1.1) with increasing OA as the relative air mass age increases, suggesting that SOA formation may have originated from biogenic precursors. Similar plots were prepared using anthropogenic precursor gases toluene and benzene (Fig. S6) showing a negative correlation with increasing organic mass concentration of r = 0.35 and a slope of -0.56. However, as the toluene and benzene concentrations are both close to the detection limit, care needs to be taken when interpreting these ratios. Generally,

although anthropogenic precursor's species are present, the VOC concentrations and trends measured suggest that the increases in OA concentrations are primarily related to biogenic emissions.

**5/ P10 lines 25-30. "plotting the two species as a function of..". Be more accurate. In fig S10, Delta MOOA and Delta LOOA are reported without defining what the delta stands for.**

We have chosen to isolate the increases in organic aerosol above the background concentrations in order to clearly represent the formation of SOA from local VOC precursors species. The background values are determined from measurements made between the airport and the valley area. During these transects, little temporal variation was observed in either the aerosol particles or in the gas phase species. The particle size distribution measured by the SMPS showed a single mode at 100 nm, concentrations of short lived VOC species such as isoprene was high, concentrations of longer lived species such as MACR+MVK were low. A full list of background values versus those in the high concentration area are listed in Table S2.

The text has been updated as follows:

During the flights, as the valley area is approached, the sampled air masses become gradually more oxidized with respect to biogenic emissions, providing us with a well-defined sample area to evaluate the contribution of biogenic SOA on background/regional air masses.  In order to isolate the formation of OA resulting from the oxidation of VOC species, the change in the OA concentrations above the background was calculated (ΔOrg). The background values were determined based on measurements during transects of the flight between the valley area and the airport. During this time, aerosol concentrations were low with little temporal variation. Particle size measurements display a single mode at 100 nm with average particle concentrations of 3000 cm$^{-3}$. Measurements of VOC species during this background period result in average concentrations of Isoprene of 1544 pptV ± 696 pptV, and lower concentrations of longer lived species MACR+MVK of 661 pptV ± 239 pptV (Table S2).

For the resolved PMF factors, LOOA and MOOA, background values were determined to be 0.27 and 0.41 µg m$^{-3}$ respectively (Table S2). Organic concentrations corrected for background concentrations are referred to as Δ-LOOA and Δ-MOOA. Plotting these two factors against the ratio of MACR+MVK+ISOPOOH/isoprene (relative air mass age) (Fig. 7 and S10), we observe a significant increase of the Δ-LOOA species with air mass age until a maximum is reached at ratios of 0.65.

**Table S2. Background and study area concentrations measured for the main species during RF20**

|  | Background | valley area |
|---|---|---|
| **Organic (µg m$^{-3}$)** | 1.05±0.39 | 2.80±0.52 |
| **MOOA** | 0.41±0.23 | 1.12±0.30 |
| **LOOA** | 0.27±0.22 | 1.39±0.21 |
| **Sulphate(µg m$^{-3}$)** | 0.04±0.02 | 0.15±0.04 |
| **Nitrate(µg m$^{-3}$)** | 0.33±0.06 | 0.69±0.14 |
| **Ammonia(µg m$^{-3}$)** | 0.11±0.11 | 0.50±0.13 |
| **Isoprene (pptV)** | 1544±696 | 962±540 |
| **MACR+MVK (pptV)** | 661±239 | 901±358 |
| **Toluene (pptV)** | 84±34 | 131±27 |
| **Benzene (pptV)** | 83±28 | 75±37 |
| **Monoterpenes (pptV)** | 201±20 | 234±34 |

**Original text: "we observe a significant increase of the LOOA species with air mass age until a maximum is reached at ratios of 0.65."**

**Here the question is what happen for higher m/z71 to m/z 69 ratio?**

Text has been added: A slower increase in concentrations of LOOA at higher ratios suggests that as the relative photochemical age of the air mass increases, LOOA becomes more oxidized and is converted to MOOA, as has recently been illustrated in chamber experiments by Palm et al., (2018). Plotting these two factors as a function of air mass age relative to anthropogenic VOC species (ratio of toluene/benzene), we observe a relatively flat and decreasing trend (Fig. S10a).

**"MOOA remains relatively stable, indicating an independent source." That's a little bit short.**

**As MOOA is associated to SO4, one can suspect that MOOA is related to long range transport episodes or regional pollution impacted by anthropogenic emissions.**

Additional text has been included.

Given that MOOA does not change with the relative air mass age in the measured area, and that it is associated with $SO_4$ and $NO_3$ species, it is reasonable to suggest that the MOOA is associated with long-range transported aerosol.

**Considering the modelling results (fig 9) it seems that anthropogenic SOA cannot be neglected (17-28% of OA, fig 9).**

Additional text has been added.

In the model, these high O:C ratios arise because of organic compounds from isoprene oxidation, which all have O:C ratio greater than 0.8, as well as some ELVOCs compounds (monomers) from monoterpenes oxidation. We can conclude from these observations that the low volatility products (ELVOCs) from monoterpenes oxidation as well as isoprene oxidation products may therefore correspond to the measured LOOA concentrations. Although the SOA contribution of anthropogenic VOC precursors is low (Couvidat et al, 2013; Sartelet et al. 2018), the results of the model show a high contribution of anthropogenic compounds (up to 30%). These anthropogenic compounds could correspond to the regionally transported SOA, potentially identified as MOOA.

**6/ P11, line15 : "we do not believe that isoprene derived SOA contributed significant amounts to the OA measured during these flights". Replace by : "we assume that….. do not…"**

**but 15-35% (fig9) is not totally insignificant.**

Yes, we meant that the contribution of isoprene-derived SOA formed via the IEPOX route is assumed to be low. The text has been updated:

"Since the measured aerosol particles are neutralized (Fig. S11) and the measured NO concentrations are still reasonable high (0.30 ppbV), we assume that isoprene derived SOA, following the IEPOX formation route, do not contribute significant amounts to the OA measured during these flights."

**7/ P12, lines 25-28: "Although isoprene emissions are much higher than those of monoterpenes and sesquiterpenes over that region"**

**Please be more quantitative**
**Additional information has been included.**

**"Isoprene derived SOA represent only about 15 to 35%"**

**15-35% is a high contribution, from my perspective, considering isoprene-SOA.**
**"4% to 7% are from semi-volatile organic compounds (pinic acid, norpinic acid and pinonaldehyde)" Please be homogenous with the legend of fig 9 :"Monoterpene products"**
**"17% to 23% from organic nitrate"**

We agree, 15-35 % is a substantial contribution. In our sentence, the word "only" was just to emphasis that this contribution was low compared to the contribution of monoterpene-derived SOA.

We understand the confusion for the reader and modified the text accordingly.

Although isoprene emissions are 2.5 times higher than those of monoterpenes and 11.6 times higher than those of sesquiterpenes over that region during the period of simulation, isoprene-derived SOA represent about 15 to 35% of the simulated OA, which is lower than the monoterpenes-derived SOA that represent 35% to 40%. Sesquiterpenes-derived SOA represents about 10%. Amongst the monoterpenes-derived SOA, 4% to 7% are monoterpene products (first generation semi-volatile organic compounds: pinic acid, norpinic acid and pinonaldehyde), 9% to 14% are ELVOCs/HOMS, and 17% to 23% are organic nitrate. In total, biogenic-derived OA represents about 66% to 80% of OA. The rest is made up by aromatics derived OA (2% to 3%) and anthropogenic intermediate and semi volatile organic compounds (17% to 31%).

**Considering this potential very high contribution of organic nitrates, they should be evidenced by the ratio NO+/NO2+**

There is little variation in the measured $NO^+/NO_2^+$ ratio. The ratio was calculated to be approximately 0.2 and was always lower than the ratio calculated during the calibration exercise (0.34).

Using the method described in Keidler-Scharr et al., (2016), we estimate the contribution of organic nitrate to be approximately 32% (of the total nitrate) for RF20 and up to 40% for RF23. However it should be noted that nitrate aerosol particles did not contribute more than 5% to the total aerosol mass. Therefore the model estimate of 17 to 23% of organic nitrate is largely overestimated. We assume that this is because hydrolysis is not taken into account in the model. Hydrolysis is said to be responsible for the elimination of the organic nitrate functionality leading to the evaporation of nitric acid from the particles (Rindelaub et al., 2016).

The text has been modified accordingly:

"The contribution of organic nitrate modeled is not reflected in the measurements, where only 5% of the total measured aerosol mass contained nitrate aerosol. This difference is likely due to hydrolysis not being accounted for in the model. However, under ambient conditions hydrolysis could eliminate the organic nitrate functionality, allowing nitric acid to evaporate from the particles (Rindelaub et al., 2016).

**8/P13, line 19 : again, 15-35% of isoprene SOA is significant.**

Original text "A lack of direct evidence of IEPOX SOA ($m/z$ 82 $C_5H_6O^+$) in the cToF-AMS measurements leads us to conclude that the formation of SOA, from isoprene precursor species was not important during this measurement period. This was confirmed through the use of the Polyphemus model, which shows that although the contribution of monoterpene and aromatic species is low compared to that of isoprene and its oxidation products, the yield of SOA formation from these precursor species is much more important."

The text has been updated.

A lack of direct evidence of IEPOX SOA ($m/z$ 82 $C_5H_6O^+$) in the cToF-AMS measurements leads us to conclude that the formation of SOA, following an IEPOX formation route from isoprene precursor species was not dominant during this measurement period. The Polyphemus model, determines a contribution of Isoprene SOA, formed through alternative pathways, of the order of 15 to 35%. Therefore, although not possible to accurately identify the formation pathway of the measured SOA, we can assume that it is at least partly associated with biogenic isoprene VOC species. The model also illustrates that although the emission of monoterpene and sesquiterpene species is low compared to that of isoprene, the yield of SOA formation from these precursor species is more important. This is in agreement with recent observations by Zhang et al., (2018), who showed that SOA is dominantly formed from monoterpene emissions in southern USA.

Zhang, H, Yee, L.D, Lee, B; H., Curtis, M.P, Worton, D.R, Isaacman-VanWertz, G., Offenberg, J. H. , Lewandowski, M., Kleindienst, T.E, Beaver, M.E, Holder, A.L., Lonneman, W.A., Docherty, K.S. , Jaoui, M, Pye, H.O.T, Hu, W, Day, D.D, Campuzano-Jost, P, Jimenez, J., Guo, H, Weber, R.J, de Gouw, J, Koss, A.R, Edgerton, E.S, Brune, W, Mohr, C, Lopez-Hilfiker, F.D, Lutz, A, Kreisberg, N.M, Spielman, S.R., Hering, S, Wilson, K.R,Thornton, J.A, Goldstein, A. H. Monoterpene SOA dominate atmospheric fine aerosol Proc.Nat Acad Sci Feb 2018, 115 (9) 2038-2043; DOI: 10.1073/pnas.1717513115

**.9/ P17 line 29. Lee et al (2016) was added in the reference list, but never cited in the text. As well, f91 is not discussed. Such discussions could bring valuable insights on the Biogenic SOA importance and origin.**

The section is updated (Page 11).

Other sources of biogenic SOA can originate from the oxidation of monoterpene and sesquiterpene VOC species, or additionally from isoprene SOA, that do not follow the IEPOX route. In all cases, the contribution of m/z 91 in the cToF-AMS mass spectra, often identified as being the $C_7H_7^+$ fragment (Lee et al. 2016; Riva et al., 2016) would been enhanced. This $m/z$ 91 was present in all OA mass spectra and was higher for the LOOA (f91 = 0.007). However, in previous studies these f91 values are considered as background (Hu et al.,2015 Lee et al., 2016), hence making it difficult to associate the measured SOA with these formation routes. It should be noted that $m/z$ 91 can also be associated with fragments of primary anthropogenic OA, and the contribution of anthropogenic aerosols from the industrial zone (Fos sur Mer) south of the flight area cannot be ruled out".

**Non-exhaustive list of technical corrections (figures/legends, mostly)**

**P2, line 4. The acronym of methacrolein is defined as MACR, but MCR is used all throughout the manuscript and in the figures (except in table 1).**

This has been changed throughout the manuscript to MACR.

**Figure 1 : add the flight numbers in the legend.**

Figure 1. Typical flight track traveling a) west (RF 15 and RF 21) and b) east (RF 20 and RF 23) of Avignon (black circle). Points of the flight track are coloured by organic aerosol concentrations.

**Figure 4 a) change the legend (see comment 3)**

This figure has been updated, please refer to our response to comment 3.

**Figure 5 : Y axis label. What is the lower cut off diameter of the SMPS 15 or 17 nm?**

[Figure]

Figure 5. Aerosol size distribution measured by the SMPS for a) RF15 b) RF20, c) RF21, d) RF23 from 17 nm up to 400 nm. The colour scale indicates aerosol concentration dN/dlogDp. Altitude is illustrated as the black line and is represented on the right hand axis.

**Figure 6 : Y axis label : add units ; figure legend : 17 or 15 nm?**

[Figure]

Figure 6. Ratios of IsopreneC/MonoterpeneC plotted as a function of the nucleation mode particles (difference between the CPC (cut off 5 nm) and the SMPS (cut off 17 nm)). Values for the four biogenic flights are included, as well as average values calculated over a number of IsopreneC/MonoterpeneC ratios (size bins of 0.5), Error bars represent ±1 σ of the average CPC5nm – SMPS values. . The black line represents the linear correlation fit.

**Figure 7 : explicit the Delta**

In response to comment no.5 text explaining the delta value has been included.

The text has been updated as follows:

During the flights, as the valley area is approached, we observe the sampled air masses become gradually more oxidized with respect to biogenic emissions, providing us with a well-defined sample area to evaluate the contribution of biogenic SOA on background/regional air masses. In order to isolate the formation of OA resulting from the oxidation of VOC species, the change in the OA concentrations above the background was calculated (ΔOrg). The background values were determined based on measurements during transects of the flight between the valley area and the airport. During this time, aerosol concentrations were low with little temporal variation. Particle size measurements display a single mode at 100 nm with average particle concentrations of 3000 $cm^{-3}$. Measurements of VOC species during this background period result in average concentrations of Isoprene of 1544 pptV ± 696 pptV, and lower concentrations of longer lived species MACR+MVK of 661 pptV ± 239 pptV (Table S2).

For the resolved PMF factors, LOOA and MOOA, the background values were 0.27 and 0.41 µg $m^{-3}$ respectively (Table S2). Organic factors corrected for background concentrations are referred to as Δ-LOOA and Δ-MOOA. Plotting these two factors against the ratio of MACR+MVK+ISOPOOH/isoprene (relative air mass age) (Fig. 7 and S10), we observe a significant increase of the Δ-LOOA species with air mass age until a maximum is reached at ratios of 0.65.

Figure 7. a) A two factor solution determined from PMF analysis of the biogenic research flights. A) i) The more oxidised organic aerosol (MOOA) associated with inorganic peaks for sulphate (red) and nitrate (blue) ,a)ii)  the less oxidised organic aerosol (LOOA) with a lower contribution of inorganic peaks. b) Variations of these two species with aging airmass (using MCR+MVK+ISOPOOH as a

proxy for photochemical age of airmass). The delta value (Δ) are calculated from the increases in OA above background concentrations (Table S10).

**Figure 9 : Put side by side all the monoterpenes derived SOA components.**

Changes have been made to the figure.

[Figure]

**Figure S3 : X axis : wrong units ! , MCR+Isoprene? / Legend : ISOPOOH is missing**

**This figure has been modified and updated.**

[Figure]

Figure S3. Vertical profiles o RH,. isoprene and its oxidation products (MVK+MACR+ISOPOOH) for: a) RF15, b) RF20, c) RF21, d) RF23.

**Figure S6 : No identified Y axis for benzene, X axis : I suppose it's Toluene/Benzene and not Benzene/Toluene**

This figure has been corrected and updated.

[Figure]

Figure S6. Comparison of total organic matter (μg m$^{-3}$) measured by the C-ToF-AMS with a) Toluene and Benzene concentrations. as a function of time. and b) against the ratio of the Toluene/Benzene.

**Figure S7 : Develop and modify the legend (you didn't calculate airmass age, just used proxies). Define Delta, flight numbers**

Figure S10 Variations of these two PMF factors with the relative air mass age calculated using either a) anthropogenic VOC or b) using MCR+MVK+ISOPOOH to calculate the relative air mass age . Δ LOOA and Δ-MOOA represent organic concentrations above the determined background (0.2 and 0.4, respectively)

**Figure S11 : Wrong legend**

This error has been corrected.

Figure S11. NH$_4$ measured to NH$_4$ predicted plotted against organic aerosol  during flight RF20. The points are coloured by SO$_4$ concentrations.